# Stratospheric Nudging And Predictable Surface Impacts (SNAPSI): A Protocol for Investigating the Role of Stratospheric Polar Vortex Disturbances in Subseasonal to Seasonal Forecasts

Peter Hitchcock[1], Amy Butler[2], Andrew Charlton-Perez[3], Chaim Garfinkel[4], Tim Stockdale[5], James Anstey[6], Dann Mitchell[7], Daniela I. V. Domeisen[8,9], Tongwen Wu[10], Yixiong Lu[10], Daniele Mastrangelo[11], Piero Malguzzi[11], Hai Lin[12], Ryan Muncaster[12], Bill Merryfield[6], Michael Sigmond[6], Baoqiang Xiang[13,14], Liwei Jia[13], Yu-Kyung Hyun[15], Jiyoung Oh[16], Damien Specq[17], Isla R. Simpson[18], Jadwiga H. Richter[18], Cory Barton[19], Jeff Knight[20], Eun-Pa Lim[21], and Harry Hendon[21]

[1]Dept. Earth and Atmospheric Sciences, Cornell University, Ithaca, NY, USA
[2]NOAA Chemical Sciences Laboratory, Boulder, CO, USA
[3]Department of Meteorology, University of Reading, UK
[4]Fredy and Nadine Herrmann Institute of Earth Sciences, The Hebrew University of Jerusalem, Jerusalem, Israel
[5]European Centre for Medium-range Weather Forecasts, Reading, UK
[6]Canadian Centre for Climate Modelling and Analysis, Environment and Climate Change Canada, Victoria, BC, Canada
[7]School of Geographical Sciences, University of Bristol, Bristol, United Kingdom
[8]University of Lausanne, Lausanne, Switzerland
[9]ETH Zürich, Zürich, Switzerland
[10]Beijing Climate Center, China Meteorological Administration, Beijing, China
[11]CNR-ISAC, Bologna, Italy
[12]Recherche en prévision numérique atmosphérique, Environment and Climate Change Canada, Dorval, QC, Canada
[13]Geophysical Fluid Dynamics Laboratory, NOAA, Princeton, NJ, USA
[14]University Corporation for Atmospheric Research, Boulder, CO, USA
[15]National Institute of Meteorological Sciences, Korea Meteorological Administration, Jeju, Korea
[16]School of Earth and Environmental Sciences, Seoul National University, Seoul, Korea
[17]Centre National de Recherches Météorologiques, Université de Toulouse, Météo-France, CNRS, Toulouse, France
[18]Climate and Global Dynamics Laboratory, National Center for Atmospheric Research, Boulder, CO, USA
[19]Space Science Division, US Naval Research Laboratory, Washington, DC, USA
[20]Hadley Centre, Met Office, Exeter, United Kingdom
[21]Bureau of Meteorology, Melbourne, Victoria, Australia

**Correspondence:** Peter Hitchcock (aph28@cornell.edu)

**Abstract.** Major disruptions of the winter season, high-latitude, stratospheric polar vortices can result in stratospheric anomalies that persist for months. These sudden stratospheric warming events are recognized as an important potential source of forecast skill for surface climate on subseasonal to seasonal timescales. Realizing this skill in operational subseasonal forecast models remains a challenge, as models must capture both the evolution of the stratospheric polar vortices in addition to their coupling to the troposphere. The processes involved in this coupling remain a topic of open research.

We present here the Stratospheric Nudging And Predictable Surface Impacts (SNAPSI) project. SNAPSI is a new model intercomparison protocol designed to study the role of the Arctic and Antarctic stratospheric polar vortex disturbances for sur-

face predictability in sub-seasonal to seasonal forecast models. Based on a set of controlled, subseasonal, ensemble forecasts of three recent events, the protocol aims to address four main scientific goals. First, to quantify the impact of improved stratospheric forecasts on near-surface forecast skill. Second, to attribute specific extreme events to stratospheric variability. Third, to assess the mechanisms by which the stratosphere influences the troposphere in the forecast models, and fourth, to investigate the wave processes that lead to the stratospheric anomalies themselves. Although not a primary focus, the experiments are furthermore expected to shed light on coupling between the tropical stratosphere and troposphere. The output requested will allow for a more detailed, process-based community analysis than has been possible with existing databases of subseasonal forecasts.

## 1 Introduction

Sudden stratospheric warmings are dramatic manifestations of dynamical variability in the polar vortices that form each winter in both hemispheres (Scherhag, 1952; Baldwin et al., 2021). They are known to lead to equatorward shifts of the tropospheric eddy-driven jets that can persist for several months (Kidston et al., 2015), and to increase the likelihood and severity of a variety of high-impact extreme events (Domeisen and Butler, 2020). Capturing these surface impacts is thus of growing concern for operational centers interested in improving their extended range forecasts on subseasonal to seasonal timescales.

A number of recent studies have explored the role of major or minor stratospheric warmings in the Subseasonal-to-Seasonal Prediction (S2S) database (Butler et al., 2019; Karpechko et al., 2018; Domeisen et al., 2020a, b; Rao et al., 2019, 2020a, b; Lee et al., 2019; Butler et al., 2020) or in individual models (Kautz et al., 2020; Knight et al., 2020; Lim et al., 2021; Noguchi et al., 2020). These studies confirm that operational models can to some extent capture the surface impacts of such stratospheric variability, and have demonstrated regionally enhanced skill at subseasonal timescales in the weeks following stratospheric warmings. However, studies based on existing databases of subseasonal forecasts are hampered by the diversity of forecast initialization dates and ensemble generation strategies, limited availability of detailed model output, and the varying ability of operational models to capture the stratospheric variability itself. Moreover, such studies must ultimately rely on correlative analyses, making causal inferences difficult to assess. Single-model studies have been extremely valuable in providing a more detailed understanding, but cannot be as robust as a controlled, multi-model intercomparison. There is a clear need at this point to more carefully evaluate and compare the relevant coupling mechanisms across operational models in order to fully exploit this important source of skill on timescales of weeks to months.

The purpose of this paper is to propose and describe a common protocol for numerical experiments to isolate and evaluate the representation of stratospheric influence on near-surface weather in subseasonal to seasonal forecast models. The intent is that by outlining and motivating a single protocol that can be adopted by multiple operational centers, such efforts can be directly compared, increasing their collective value.

The protocol presented here is primarily based on a zonally-symmetric nudging technique that has been used successfully to identify stratospheric influences on the tropospheric circulation in both hemispheres (Simpson et al., 2011; Hitchcock and Simpson, 2014; Zhang et al., 2018; Jiménez-Esteve and Domeisen, 2020). In essence, by comparing an ensemble hindcast in

which the stratosphere is constrained to the observed evolution, to a second hindcast in which the stratospheric circulation is constrained to climatology, the tropospheric impacts of the stratospheric anomalies can be isolated. Moreover, the protocol will request a more complete set of output than is typically available from existing databases of subseasonal forecasts. The requested variables are relevant to understanding both the coupling processes and the surface impacts themselves. The experiments described here will thus represent a significant step forward from the previous intercomparisons of operational forecasts both by allowing deeper investigations into the relevant coupling processes, and by removing the confounding influence of differences in stratospheric forecast skill.

Although the zonally-symmetric nudging is related to other nudging approaches (Jia et al., 2017; Kautz et al., 2020; Knight et al., 2020), in this case stratospheric circulation anomalies are imposed through a linear relaxation term that acts only on the zonally symmetric component of the stratospheric circulation. The purpose of this technique is to permit eddies to vary in a dynamically consistent way across the tropopause. This is particularly relevant for the planetary waves that play a central role coupling the stratosphere and troposphere. This approach has been shown theoretically (Hitchcock and Haynes, 2014) and practically (Hitchcock and Simpson, 2014) to avoid any significant artifacts. The latter work has also shown that much of the surface response to SSWs is obtained by nudging the zonal mean state alone. Nonetheless, it is possible that there are aspects of the surface response that are related to zonal asymmetries in the stratosphere, and experiments in which the full stratospheric state is nudged are also described and requested at a secondary priority level. This choice is further discussed below.

While the protocol as outlined is intended to be applicable to any stratospheric event of interest, we suggest that it be initially applied to three specific recent events: the boreal sudden warmings that occurred in February 2018 and January 2019, and the austral sudden warming that occurred in September 2019. Each of these was followed by surface extremes that studies have suggested arise in part because of the stratospheric event.

This project is coordinated by the Stratospheric Network for the Assessment of Predictability (SNAP) working group that is a joint activity of the World Climate Research Programme (WCRP) Stratosphere-troposphere Processes and their Role in Climate (SPARC) project and of the Subseasonal-to-Seasonal Prediction (S2S) project that is supported by both the WCRP and the World Weather Research Programme (WWRP).

This paper describes the overall experimental design as well as details of the nudging approach. Data produced by this project will be made available to the community through the Centre for Environmental Data Analysis (CEDA), with the aim of providing researchers with a resource to investigate the dynamics of stratosphere-troposphere coupling. While not a central goal of the experiments, the case studies span periods with several distinct phases of the quasi-biennial oscillation (QBO) and the occurrence of several large-amplitude Madden-Julian Oscillation (MJO) events. As such these experiments are expected to be valuable to several other SPARC and S2S projects, including the S2S MJO and Teleconnections group, the QBO initiative (QBOi), and Stratospheric And Tropospheric Influences On Tropical Convective Systems (SATIO-TCS).

The paper is outlined as follows. The next section describes four specific goals that the proposed experiments are intended to achieve. The third section describes in detail the general experimental protocol that can be applied to study any stratospheric event of interest, and specifies details of the nudging, including the reference states towards which the nudged experiments are relaxed. The trade-offs of different nudging approaches are also discussed further. In the fourth section the three target events

of interest are described in further detail. The fifth section lays out the model output requested from the forecasts, and the final section includes a list of participating models and a brief concluding outlook.

## 2  Overview and Motivations

The basic experimental design proposes to focus on the evolution of specific events of interest, using the following sets of forecast ensembles:

**free**  A standard forecast ensemble in which the atmosphere evolves freely after initialization. The method of initialization and of generating ensemble members is not specified and can be determined by the participating modeling groups.

**nudged**  A nudged ensemble in which the zonally symmetric stratospheric state is nudged globally to the observed time evolution of the stratospheric event of interest.

**control**  A nudged control ensemble in which the zonally symmetric stratospheric state is nudged globally to a time-evolving climatological state.

As discussed in the introduction, the **free** ensembles along with the zonally symmetric **nudged** and **control** ensembles are of highest priority. However, zonally symmetric nudging can be difficult to implement in models with grids that are not aligned along fixed latitudes. Thus two additional ensembles are requested at lower priority:

**nudged-full**  A nudged ensemble in which the full stratospheric state (including zonally asymmetric components) is nudged globally to the observed time evolution of the stratospheric event of interest.

**control-full**  A nudged control ensemble in which the full stratospheric state (including zonally asymmetric components) is nudged globally to a time-evolving climatological state.

The reference states for the **nudged** and **control** ensembles are computed from ERA5 reanalysis output (Hersbach et al., 2020) on native model levels. These coincide with isobaric surfaces at the stratospheric levels where the nudging is applied. Details of how the climatological state is computed are given in the Methods section below.

The protocol targets forecast integrations of 45 days, and an ensemble size of 50 to 100 members. For each of the three case studies, two specific initialization dates for each type of integration are proposed; these are discussed in the context of the specific target events described in Section 3.

The impact of the stratosphere on the troposphere can be confounded by unrelated dynamical variability within the troposphere. Hence the choice to emphasize ensemble size over the number of initialization dates is intended to allow for statistically and dynamically meaningful comparisons of these specific events across participating models.

There are four central motivations for the proposed forecast experiments, presented in the following subsections. In addition, these experiments are expected to provide useful insights into coupling between the tropical stratosphere and troposphere. These secondary motivations are discussed following the four primary goals.

## 2.1 Quantify stratospheric contributions to surface predictability

Through nudging the stratosphere to observations, the **nudged** ensemble will provide a 'perfect' forecast of the stratosphere's zonal mean state. The forecast skill attained can be compared to that attained by the **control** ensemble (amounting to a 'climatological' stratospheric forecast) and the **free** ensemble to quantify the contribution of a successful forecast of the stratosphere. These experiments will provide a multi-model assessment of the potential increase in skill associated with an improved representation of the stratospheric state, and an up to date assessment of the present skill that is achieved by each model.

Many authors have noted that the surface response seen in the composite average following SSWs is not seen in every individual event. An important question here is whether this inter-event diversity is predictable to some extent on S2S timescales. For instance, if the equatorward shift of the North Atlantic jet following SSWs depends on the state of the MJO, this modulation of the response may be predictable in advance. As a second example, a recent study by Dai and Hitchcock (2021) suggests that the North Pacific response following SSWs depends strongly on the nature of SST anomalies at the onset of the stratospheric event. This suite of experiments will allow a case-by-case assessment of S2S predictability arising from the stratospheric state by including multiple case studies of interest.

## 2.2 Attribute extreme events to stratospheric variability

The proposed protocol will also provide a means of assessing or formally attributing the contribution of the stratosphere to an extreme event of interest (Domeisen and Butler, 2020). Extremes that have been associated with sudden warmings in recent years include cold air outbreaks in the Northern Hemisphere (Kolstad et al., 2010; Afargan-Gerstman et al., 2020; Huang et al., 2021; Charlton-Perez et al., 2021) and hot, dry extremes over Australia (Lim et al., 2019). This goal is closely related to the growing sub-discipline that focuses on attributing the occurrence of particular extremes to climate change and variability (National Academies of Sciences, Engineering, and Medicine, 2016).

Consider some extreme event $A$ that is thought to have been associated with a specific sudden stratospheric warming (SSW), for instance the cold air outbreak (CAO) that occurred in Europe following the sudden warming in February 2018. The probability of such an event occurring $p_0 = p(A)$ might be estimated from the observed climatological frequency of similar events, or from a set of forecasts that sufficiently represent the variability of the climate system from a given subseasonal forecast model, or from a combination of both (Sippel et al., 2015). Given the **nudged** ensemble, one can then estimate the probability of a similar event occurring given the weakened state of the stratospheric polar vortex $p_1 = p(A|V^-)$. The Relative Risk (see, e.g. Paciorek et al., 2018) of this CAO might then be calculated as RR = $p_1/p_0$. Relative Risk values of RR $> 1$ would then imply an increased risk of a CAO under a weakened vortex state, whereas RR $< 1$ would imply the opposite. This can also be compared to the probability of such an event occurring in the counterfactual situation that the sudden warming did not occur, $p_0' = p(A|V^0)$, computed from the **control** ensemble, allowing further for the calculation of necessary or sufficient causation probabilities (Hannart et al., 2016). In the context used here, the RR is the most appropriate measure of risk because the data are likely to be non-Gaussian (Christiansen, 2015).

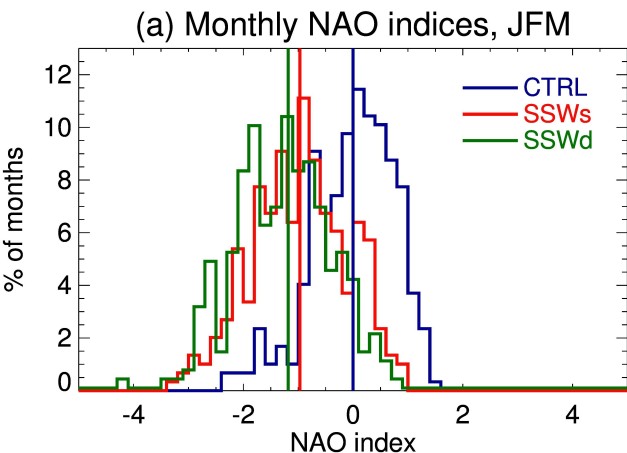

**Figure 1.** Monthly mean NAO indices from a set of nudged integrations similar to those described in this protocol. The zonally symmetric component of the stratosphere in the CTRL run is nudged to the model climatology, while those of SSWs and SSWd are nudged, respectively, to the evolution of a vortex split and a vortex displacement event simulated by a free-running configuration of the same model. (Fig. 12a from Hitchcock and Simpson, 2014, copyright American Meteorological Society. Used with permission.)

As an example, Fig. 1 shows monthly mean indices of the North Atlantic Oscillation (NAO) (from Hitchcock and Simpson, 2014). The probability of occurrence of a strongly negative monthly mean NAO state is much more likely in the aftermath of a sudden stratospheric warming than under a 'counterfactual' scenario during which the stratosphere was close to its climatological state. This result was robust to nudging the zonal mean state to two different reference events (labelled 'SSWs' and
'SSWd') taken from the free running version of the model.

The interpretation of the Relative Risk becomes more challenging in a forecast context, since the probability of an extreme event is strongly conditional on the initial conditions known at the time of the forecast. As the forecast initialization date grows closer to the event of interest, the forecast ensembles will begin to forecast the event with increasing fidelity; that is, the probability of occurrence conditional on initial conditions $n$ days prior to an event, $p(A|IC(n))$ will grow.
One practical way to frame this question is to ask whether improving the forecast of the stratospheric state can lead to earlier accurate forecasts of the event in question. Alternately, one may ask whether degrading the forecast of the stratosphere leads to degraded forecasts of the event. Both framings are enabled by the proposed experiments.

Figure 2 provides an example from NCEP CFSv2 monthly forecasts (Saha et al., 2014) of March 2018 temperatures over Europe for different initialization dates. A sudden stratospheric warming occurred on 12 February 2018. The forecast model
did not predict the SSW with any certainty until the initializations in the 1-10 February period. There is a significant change in the March surface temperatures over Europe for initializations before and after the stratospheric event was captured in the prediction system, with forecasts initialized with the SSW information more closely capturing the observed March temperatures. But do these differences arise solely because the forecast model finally captured the SSW, or because the lead-time had decreased? With the three experiments proposed and applying this to multiple initializations before the event, it would be clear

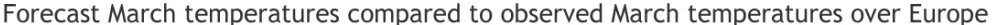

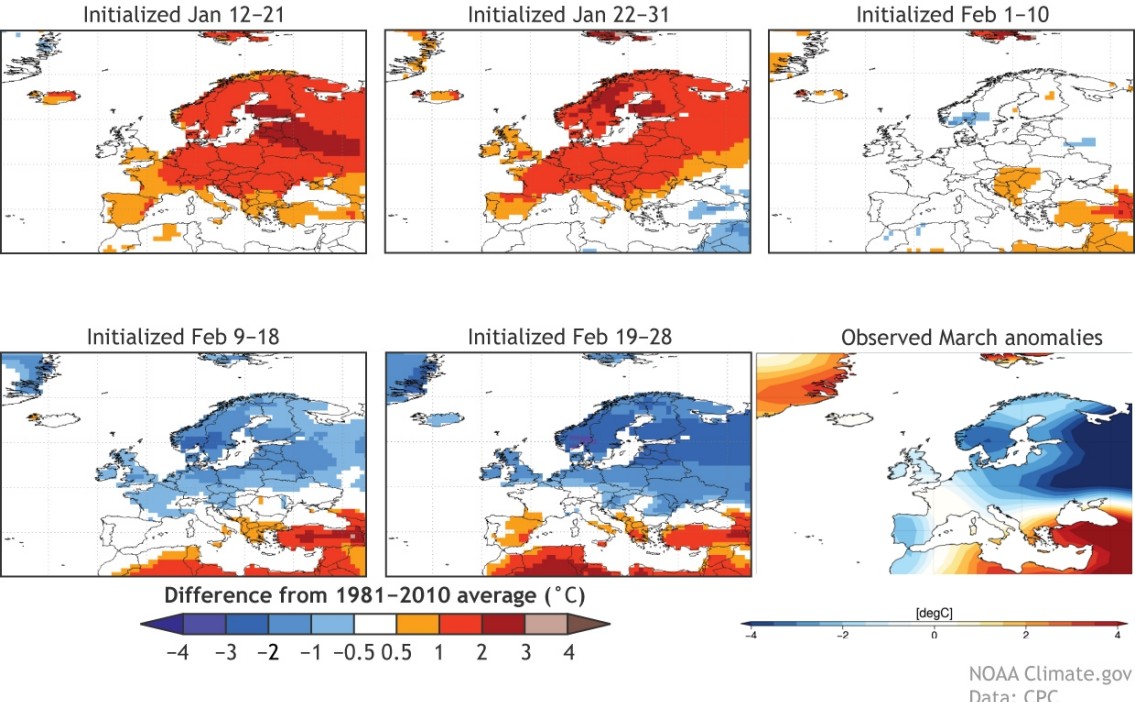

**Figure 2.** Forecast and observed monthly averaged temperature anomalies over land for March 2018 over Europe. Forecasts are from NCEP CFSv2, averaged over a total of 40 ensemble members, four of which are initialized on each date in the range of initial dates given in the captions. Observations are from the NCEP/NCAR reanalysis.

whether or not having the 'perfect' stratosphere (**nudged** or **nudged-full** ensemble) for runs initialized in mid-January would have given more accurate forecasts at longer leads.

A very similar approach has been adopted by Kautz et al. (2020) who made the distinction between 'probabilistic' and 'deterministic' forecasts of the extreme event in question. They presented evidence from the ECMWF model that a perfect forecast of the stratospheric anomalies in early 2018 would increase the predicted odds of extreme cold weather over Europe from ~5% to ~45%. These odds then increase further as forecasts are made closer to the event.

The common and comparable set of integrations from a range of operational centers made available by this project will allow this finding to be extended to other extreme events and will allow further development of this methodology. We aim to have a large enough ensemble size to allow for the direct study of well-constrained extreme events, but if necessary may also supplement the model data with extreme value statistics (e.g. Sippel et al., 2015). Ensemble sizes of 50-100 are sufficient to understand large-spatial scale and persistent (weekly) extremes, such as cold or NAO events, but the latter method will be required for 'noisier' fields such as precipitation.

## 2.3 Assess the mechanisms underlying stratospheric coupling in individual models

Imposing stratospheric anomalies through a nudging procedure has been shown to significantly impact the near surface flow (e.g. Douville, 2009), even if only the zonally symmetric component is imposed (e.g. Simpson et al., 2011; Hitchcock and Simpson, 2014; Zhang et al., 2018; Jiménez-Esteve and Domeisen, 2020). By comparing the difference between the **nudged** and **control** ensembles, the processes that drive this downward coupling can be diagnosed in each model for a variety of events of interest. It is of particular interest to better understand why some specific stratospheric events are followed by the 'canonical' equatorward shift of the tropospheric eddy driven jets, while others are not. The two boreal and one austral case studies proposed were followed by a diversity of tropospheric responses, including two cases which exhibited the 'canonical' response (the 2018 boreal and 2019 austral cases) and one which did not (the 2019 boreal case). This set of experiments will thus shed light on whether these diverse responses were determined by stratospheric causes, or whether they are determined by competing effects such as tropical tropospheric variability or independent mid-latitude dynamical processes (e.g. Knight et al., 2020). In either case, the statistical sampling afforded by a multi-model set of forecast ensembles with detailed diagnostics will allow for new and deeper insights into the mechanisms responsible for the tropospheric response. Moreover, each event also coincided with specific surface extremes that produced significant societal impacts. This set of experiments will provide quantitative insight into the mechanisms responsible for these surface extremes. The data request has been designed to allow for a more detailed analysis of these processes than has been possible with existing databases of subseasonal forecasts. Ultimately this understanding will help both future operational system design and practical use of subseasonal forecasts.

## 2.4 Quantify the role of the stratosphere in upward wave propagation

The onset of a sudden stratospheric warming is marked by the reversal of the climatologically westerly zonal mean zonal winds in the mid stratosphere. Operational forecasts can, on average, successfully forecast this reversal starting about two weeks prior, but this depends strongly on the specifics of the event in question (Tripathi et al., 2015; Domeisen et al., 2020a; Rao et al., 2020a, b). A key issue is the successful forecasting of the rapid growth in planetary-scale Rossby waves that drives the breakdown of the stratospheric polar vortex. This requires capturing both tropospheric precursors for these waves (e.g. Garfinkel et al., 2010), as well as their interaction with the stratospheric flow (e.g. Hitchcock and Haynes, 2016; de la Cámara et al., 2018; Lim et al., 2021; Weinberger et al., 2021).

A fourth goal for this protocol is to determine how well forecast systems capture this initial amplification of planetary waves. In particular, the first of the initialization dates has been chosen just prior to the periods of enhanced wave driving that led to the breakdown of the stratospheric polar vortex (as discussed in section 4 below). By comparing the evolution of the wave field in the **control** and **nudged** ensembles, the role of the stratospheric state in determining the wave amplification can be isolated and compared with the importance of capturing specific tropospheric precursors. This will reveal how well forecast models can predict the evolution of the planetary waves on a given zonally symmetric background, allowing for quantitative intercomparison. Further comparison with the **free** ensemble will provide detailed insight into the ability of individual models to forecast the complex interactions responsible for the amplification of the wave field.

## 2.5 Secondary Science Questions

Although the emphasis in the design of SNAPSI has been on extratropical coupling between the stratosphere and troposphere, the experiments are expected to provide further insights into coupling between the tropical stratosphere and troposphere, and between the tropics and extratropics in both the troposphere and stratosphere. We outline in this section several potential questions that may be addressed with these experiments.

### 2.5.1 Representation of the Quasibiennial Oscillation

These experiments may be useful for examining the model representation of the QBO. Since the QBO is a nonlinear oscillation driven by wave-mean flow interactions, the waves and the mean flow are tightly coupled: the waves influence the evolution of the mean flow, and vice versa. In the **nudged** ensemble, upward-propagating equatorial waves that force the QBO (both resolved and parameterized) will encounter essentially identical zonal-mean zonal wind profiles in all models. This allows wave forcing to be directly compared between models absent the complication of differing background zonal-mean winds. This approach has been used previously to assess the response of wave forcing to changing vertical resolution in a single model (Anstey et al., 2016). In the **free** experiment, equatorial winds can respond to the wave forcing. To the extent that model biases have time to develop over the 45-day hindcast period, results from the **nudged** ensemble may yield insight into the origin of biases in the **free** runs. In particular, current QBO-resolving models are typically unable to maintain realistic QBO amplitude in the lowermost tropical stratosphere (Stockdale et al., 2020; Richter et al., 2020), and some S2S models lose nearly the entire QBO signal within a typical 40-day reforecast (Garfinkel et al., 2018).

### 2.5.2 Stratospheric influences on tropical convection

Recent work has highlighted a variety of potential stratospheric impacts on organized tropical convection (Haynes et al., 2021). Notably, the phase of the QBO has been shown to have a significant impact on the strength and persistence of the MJO (Son et al., 2017). This impact has an apparent effect on the predictive skill of the MJO, in that forecasts of the MJO remain skillful at longer lead times during the easterly phase of the QBO (Martin et al., 2021b). Sudden stratospheric warmings have also been shown to shift and enhance regions of tropical convection (Kodera, 2006; Noguchi et al., 2020). A wide range of coupling pathways and mechanisms have been proposed, but fundamental understanding remains limited, in part due to the large scale separation between the planetary scales of the stratospheric variability and the mesoscale to synoptic scale of tropical convection (Haynes et al., 2021).

Imposing stratospheric variability through nudging as is proposed here can isolate the importance of the stratospheric state on convection in the forecast models. Nudging techniques similar to those adopted by SNAPSI have been used to study both tropical (Martin et al., 2021a) and extratropical (Noguchi et al., 2020) pathways in single-model contexts; SNAPSI will allow for a multi-model investigation of these effects.

### 2.5.3 Stratospheric pathways for teleconnections

The stratosphere is thought to modulate the remote impacts of a variety of climate drivers, spanning from shorter time-scale blocking events and seasonal snow-cover anomalies to El Niño-Southern Oscillation (ENSO) and sources of decadal variability such as the solar cycle (for a more complete list, see Butler et al., 2019). Thus the stratosphere may play an important role in correctly capturing the response to a broad range of subseasonal predictors. In many cases the detailed mechanisms responsible for the modulation remains an open area of study. To the extent that these teleconnections depend on the zonal mean state of the stratosphere, comparisons between the **nudged** and **control** ensembles will provide a clear means of assessing the stratospheric pathway at play for those teleconnections that are active during the selected case studies. In such cases they should be active in the former ensemble, but absent in the latter. An analysis of the teleconnection in the **free** ensemble may then provide an assessment of the skill of each model in capturing the relevant pathway. It may also yield insight into specific model biases or deficiencies that prevent skill arising from the stratospheric pathways from being realized.

## 3 Methodology

### 3.1 Reference states

The reference states (Hitchcock, 2022) have been prepared from the ERA5 reanalysis (Hersbach et al., 2020). The zonally symmetric reference states used for the **nudged** and **control** ensembles are the instantaneous zonal mean temperature and zonal wind output from ERA5 at the native 137 model levels at six hourly intervals, interpolated to a 1 degree horizontal grid. No relaxation is imposed on the meridional (or vertical) winds in these two ensembles. For the full ensembles which require zonally varying information, the zonal and meridional wind fields are provided along with the temperatures, again on a 1 degree horizontal grid at the native vertical resolution of ERA5.

The climatological state for the **control** ensemble is computed based on the 40-year period from 00:00 UTC 1 July 1979 to 18:00 UTC 30 June 2019. Leap years are handled by using the 365 consecutive days following 1 July, omitting 30 June; thus 29 February is treated as 1 March for leap years. Thus any discontinuities from the end points of the climatology or from omitting leap days are introduced between 30 June and 1 July, outside forecast periods of interest. The climatological state is then further smoothed in time by a 121-point (30-day) triangular filter to reduce residual high-frequency features from the limited sampling of the climatological state. A further modification to the reference state for the **control** ensemble is discussed in the next section.

### 3.2 Nudging specification and reference states

'Nudging' specific components of the atmospheric circulation by means of an artificially imposed relaxation to a given state has been used by many studies as a means of testing dynamical hypotheses. However, the introduction of an artificial linear relaxation into the equations of motion can produce unintended consequences (e.g. Shepherd et al., 1996; Hitchcock and Haynes, 2014; Orbe et al., 2017; Chrysanthou et al., 2019).

This section describes in detail the nature of the nudging relaxation to be used in this protocol, which is designed to avoid such consequences. The intent for the **nudged** and **control** ensembles is to prescribe the zonally-symmetric component of the stratospheric flow without indirectly constraining the troposphere or affecting the planetary waves that play a central role in the coupling between the two. The nudging is specified as a relaxation tendency of the form $-\tau^{-1}(X - X_r)$, where $X$ is either the zonal mean temperature or zonal wind, and $X_r$ is the zonally symmetric reference state to which the flow is constrained. The nudging tendency is imposed equally on all longitudes (at a given latitude and height), to avoid directly affecting the wave field. The timescale of the nudging varies with pressure, tapering gradually from infinite (i.e. no nudging) below a lower limit of $p_b = 90$ hPa, to full strength at $p_t = 50$ hPa, following a cubic profile $\left(\frac{p_b - p}{p_b - p_t}\right)^3$. At full strength the nudging timescale is 6 hours. The nudging is to be imposed at all latitudes equally.

The choice of levels to nudge was made to constrain as much of the stratosphere as was feasible without directly impacting the troposphere. The impacts of imposing nudging at different levels has been considered in detail in a simpler model context by Hitchcock and Haynes (2016), who found that the surface impacts were stronger when the lower stratosphere was better constrained. The lower stratosphere has also been shown to be particularly relevant for understanding the impacts of SSWs in the context of more comprehensive models and observations (e.g., Hitchcock et al., 2013; Maycock and Hitchcock, 2015; Karpechko et al., 2017). The choice to ramp up the nudging from 90 hPa to full strength at 50 hPa is similar to the lowest level of nudging considered by Hitchcock and Haynes (2016), remains well above the level of the extratropical tropopause (which is more than a scale height below), and is low enough to constrain the tropical lower stratosphere as well.

While the nudging profile is specified in pressure coordinates, the intent is for the nudging strength to be constant on model levels and can be converted using 'typical' pressures appropriate for the details of the vertical coordinate system of a given model.

For the **nudged-full** and **control-full** ensembles, the same nudging profile is applied to the horizontal winds and temperature.

For the **control** ensemble, the protocol specifies nudging the zonally symmetric components of the stratosphere towards the climatology. Since the initial conditions are in some cases some ways away from the climatology, nudging at full strength to the climatology will generate undesirable transients as the stratosphere adjusts towards the climatological state.

In order to reduce this initial shock, the reference state for the **control** ensemble is interpolated smoothly from the observed evolution to the climatology over the first 5 days of the forecast period. For instance, the temperatures is relaxed towards a state $T_r$ defined by

$$T_r(t) = T_o(t)\left(1 - f(t - t_i)\right) + T_c(t)f(t - t_i)$$

where $T_o$ is the instantaneous reference state, $T_c$ is the reference climatology computed as described in the previous section, $t$ is the time, $t_i$ is the starting time of the forecast, and $f$ is an interpolating function given by

$$f(t - t_i) = \begin{cases} 0 & \text{if } t - t_i < 0 \\ \sin^2\left(\frac{\pi}{2}\frac{(t - t_i)}{\Delta t}\right) & \text{if } 0 \leq t - t_i < \Delta t \\ 1 & \text{if } t - t_i \geq \Delta t \end{cases}$$

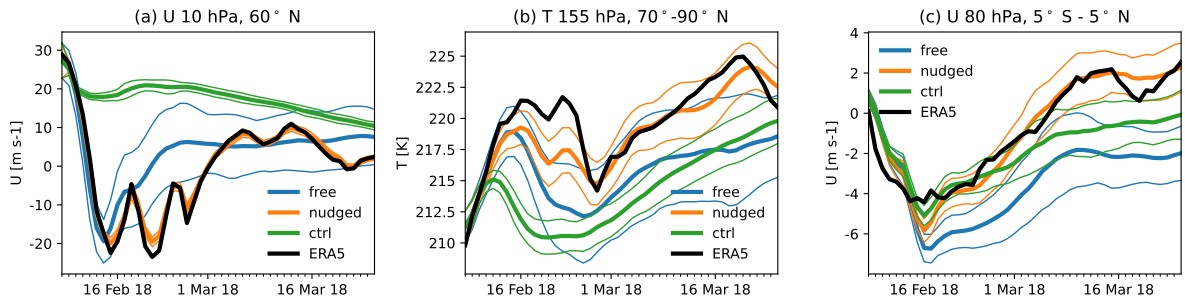

**Figure 3.** Evolution of zonal mean U and T in several regions indicated in the captions for **free**, **nudged**, and **control** forecast ensembles produced by the CESM2 model. Forecasts are initialized 12 Feb 2018; the forecast ensembles consist of 50 members. The thick lines indicate the ensemble mean, the thin lines indicate the ensemble spread (one standard deviation). Solid black lines show ERA5 output.

with $\Delta t = 5$ days. A similar adjustment should be adopted for the **control-full** ensemble.

Figure 3 demonstrates the effects of zonally symmetric nudging in forecast ensembles generated by the CESM2 model (see below for model references). In the mid-stratosphere (Fig. 3a), the nudging successfully constrains the zonal mean zonal wind in the **nudged** ensemble to follow the rapid deceleration of the winds in reference state, capturing in particular several

fluctuations of the wind following the initial reversal that were not captured by the **free** forecast. Residual differences between the nudged forecast and the ERA5 reference state arise from issues of vertical interpolation. In contrast the **control** run exhibits a gradual weakening of the climatological westerlies after an initial period during which the reference state is transitioning to the climatology. Both **nudged** and **control** exhibit much weaker ensemble spread. The nudging also constrains the zonal mean state in the lower stratosphere, well below the nudging region (Fig. 3b). Again the **nudged** ensemble better captures the gradual

warming at these levels than does the **free** ensemble, and the spread in both **nudged** and **control** is substantially smaller than in the **free** forecast. Finally, the nudging is also able to improve the zonal mean zonal winds in the tropical lower stratosphere (Fig. 3c).

Figure 4 quantifies the effects of the nudging on the zonal mean winds and meridional heat fluxes in **free**, **nudged**, and **control** ensemble forecasts from the CESM2 model. Fig. 4a-c show the ensemble mean (contour lines) and spread (filled

contours) in the zonal mean zonal wind, averaged over forecast days 10 through 40. Despite being averaged over a month following the event, the Arctic polar vortex is much weaker in the **free** and **nudged** runs than in the **control** ensemble. The ensemble spread in **nudged** and **control** is much reduced within the nudging region relative to that of the **free** ensemble, as intended.

The nudging specification in the **nudged** and **control** runs is, by contrast, not intended to directly impact the zonally asym-

metric component of the flow. The statistics of the planetary waves in particular are found to be comparable to the free running case (Fig. 4d-f), though the ensemble spread increases with overall magnitude of the meridional heat flux in the **control** run given the stronger vortex on which the waves can propagate. One exception is that wave amplitudes in the upper stratosphere can grow larger in the presence of nudging; this is in part because the nudging prevents the wave transience from decelerating

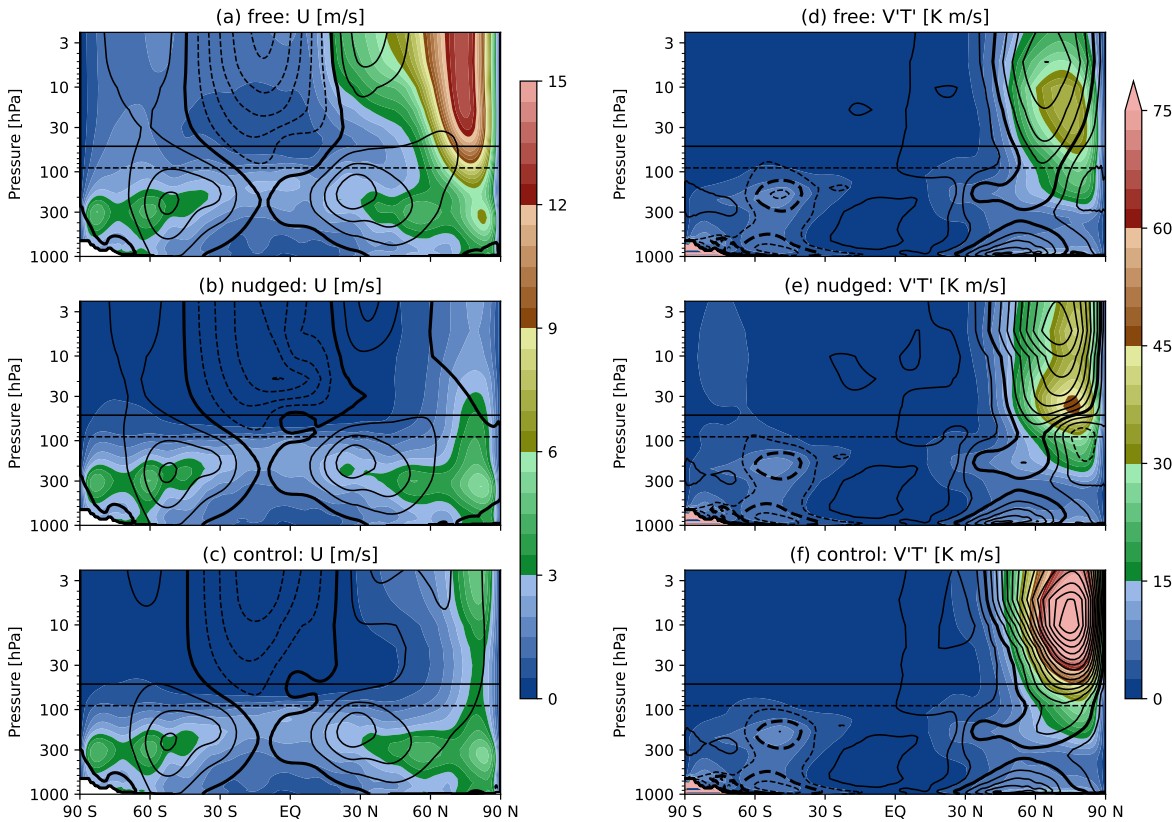

**Figure 4.** Ensemble mean (contour lines) and spread (shading) of zonal mean (a-c) zonal wind and (d-f) meridional heat flux in CESM2 forecasts initialized 12 Feb 2018, averaged over forecast days 10 to 40. The contour interval for the zonal wind is 10 m/s, for the meridional heat flux it is 5 K m/s. The forecast ensemble in (a,d) is **free** running, while those in (b,e) and (c,f) correspond to the **nudged** and **control** integrations. The horizontal lines indicate the lower (dashed) and upper (solid) bounds of the nudging transition region; above the latter the nudging is imposed at full strength.

the mean flow, allowing planetary waves to propagate higher before they encounter critical levels. In some cases this can result in unusually strong winds in the upper stratosphere and lower mesosphere; however this is not expected to influence the evolution of the lower stratosphere or its interactions with the troposphere.

Because the wave field is not directly controlled by the nudging, the zonal mean forcing produced by the internally-generated wave field can differ substantially from that consistent with the evolution of the reference state, particularly for the **control** ensemble. Since the meridional circulation is largely determined by the forcing associated with the waves (e.g. Plumb, 1982; Haynes et al., 1991), misrepresentation of the wave field can result in spurious meridional circulations and the potential for unintended remote effects. However, Hitchcock and Haynes (2014) has shown that the spurious circulations are largely confined to within the region of nudging, while the non-local circulation below the region of the nudging associated with 'downward control' is to a close approximation consistent with the forcings that produced the reference state. This implies that any down-

ward influence associated with these circulations can be expected to be present in the **nudged** ensemble, and absent in the **control** ensemble. Spurious circulations within the nudging region may give rise to anomalous transport of constituents within the stratosphere, but this is not expected to be of concern on the subseasonal timescales relevant to the present protocol.

The presence of a nudging layer can also give rise to a 'sponge-layer feedback' like response (Shepherd et al., 1996), which is characterized by spurious zonal mean temperature and wind anomalies generated just below the layer of nudging in response to tropospheric torques that differ from the reference state. These effects have also been shown to be negligible on these timescales (Hitchcock and Haynes, 2014).

## 3.3   Zonally-symmetric versus full-field nudging

This protocol emphasizes the use of a zonally-symmetric nudging approach, although additional ensembles with full-field nudging are also requested at a secondary priority level. This section discusses some of the primary merits and drawbacks to each approach, and provides justification for this prioritization.

There are two primary arguments for full-field nudging: one scientific and one practical. First, there may be some important role for stratospheric asymmetries in determining the tropospheric response to SSWs and thus the full-field nudging may represent an upper-bound of improved forecast skill arising from a perfect stratospheric forecast. Second, implementing zonally-symmetric nudging is technically difficult and computationally expensive in models operating on a grid that is not aligned with the parallels; full-field nudging may thus make it easier for more models to carry out the protocol.

There are also two primary arguments for zonally-symmetric nudging. First, by leaving the wave field to evolve freely, the experiments will allow us to investigate the impact of stratospheric mean-state biases on the forecast of the planetary wave field. Second, that we have a deeper theoretical understanding of the consequences of zonal mean nudging.

Past work (e.g. Hitchcock and Simpson, 2014) has demonstrated that much of the surface response is in fact captured by the zonal mean nudging approach alone. Planetary waves in the extratropical stratosphere are suppressed following SSWs, which means zonal asymmetries in the stratosphere are weak. This work emphasized the time-mean jet shift component of the response rather than the shift in probability of extremes such as cold air outbreaks, so it is possible that the zonal mean nudging will nonetheless miss some possible impacts arising from the asymmetric component of the stratosphere.

It is not clear, however, that full-field nudging will really provide an upper bound on the downward impacts of the stratosphere. Nudging of any kind in the presence of strong balance constraints implies that there will be unintended, remote consequences of including an artificial forcing. For instance, nudging the asymmetric component of the stratosphere will introduce an effective and highly artificial reflecting layer for any large-scale Rossby waves that are not consistent between the model forecast and the nudged reference state. The induced zonal asymmetries in the stratosphere may also act as an effective stratospheric source of waves. These may produce unintended biases in the surface flow; these effects have not been quantified. In contrast, the dynamical artifacts associated with zonally symmetric nudging are better understood (Hitchcock and Haynes, 2014).

In summary, there are strong arguments for carrying out both symmetric and full-field nudging forecasts. Both are thus included in the protocol, and we expect that enough modeling centers will carry out both in order to carry out some meaningful

**Table 1.** Case studies and forecast initialization dates. The **nudged-full** and **control-full** ensembles are requested only for the later of the two initialization dates for a given event.

| (Hemisphere) Event | Initialization Dates | |
|---|---|---|
| (NH) 12 Feb 2018 | 25 Jan 2018 | 8 Feb 2018 |
| (NH) 2 Jan 2019 | 13 Dec 2018 | 8 Jan 2019 |
| (SH) 18 Sep 2019 | 29 Aug 2019 | 1 Oct 2019 |

comparisons between the two approaches. On balance, however, the additional science questions that can be addressed with the symmetric nudging approach was considered worth prioritizing.

## 4 Case Studies

The ensemble forecasts just described will be applied to three recent events: the major sudden stratospheric warmings of 2018 and 2019 in the Northern Hemisphere, and the near-major warming of 2019 in the Southern Hemisphere. This section reviews the evolution of these three events, highlighting the evolution of the stratospheric polar vortex and the response of the tropospheric Northern and Southern annular modes (NAM and SAM, respectively) along with the closely related NAO and Arctic Oscillation (AO). Notable high-impact events that may be related to the stratospheric anomalies are also discussed. Finally, the state of other modes of climate variability are briefly summarized, including the QBO, ENSO, and the MJO. These remote climate drivers are thought to be relevant both to the stratospheric evolution and potentially to the surface impact; their state may thus be relevant for interpreting the surface impacts found in the forecast ensembles.

Two initialization dates are requested for each event (Table 1). One date is chosen about three weeks prior to the surface extreme of interest, in order to identify the contribution of the stratosphere to its forecast on subseasonal timescales (motivations 1, 2, and 3). A second date is chosen prior to the onset of the stratospheric warming in order to assess the representation of the onset of the event (motivations 1, 3, and 4). The former has higher priority than the latter, although they are listed chronologically in Table 1. Thursdays are chosen since nearly all models that contributed to the S2S database contributed forecasts initialized on Thursdays, making it easier to compare the two datasets. Further justification for the initialization dates selected are provided in the case-by-case discussion below.

### 4.1 Boreal Major Warming of 12 February 2018

The Arctic polar vortex split in early February of 2018, leading to a reversal of the zonal mean zonal wind at 60° N, 10 hPa on 12 February 2018. Prior to the event (Fig.5) the vortex was near to its climatological strength; it weakened rapidly throughout the depth of the stratosphere, coincident with large-amplitude vertical fluxes of wavenumber-two wave activity. Lower stratospheric anomalies persisted into late March of 2018.

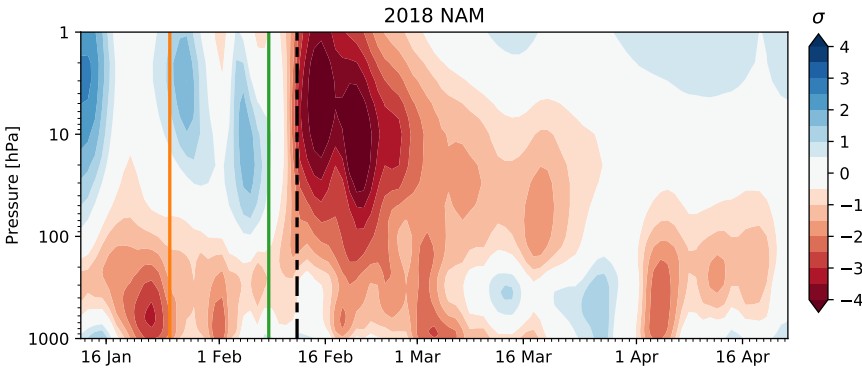

**Figure 5.** NAM indices during the February 2018 boreal major warming. The NAM indices are computed from ERA5 geopotential height anomalies following the methodology of Gerber and Martineau (2018). The vertical black dash-dotted line indicates the date of the wind reversal at 10 hPa, 60° N. The green and orange vertical lines indicate the requested initialization dates.

The tropospheric NAM responded strongly to these stratospheric anomalies, exhibiting a shift to negative values from mid-February through mid-March, consistent with the composite mean response to sudden stratospheric warmings. The NAO index was strongly negative in late February, coinciding with unusually cold weather over much of Europe and Asia during the last two weeks of Feb. (Lü et al., 2020), bringing, for example, snow to Rome and several notable winter storms to the UK.
Precipitation patterns also shifted, bringing persistent rain to the Iberian peninsula, ending an extended period of drought (Ayarzagüena et al., 2018).

Of the three proposed case studies, this first case has been the most actively studied to date. In a study of the S2S database, Rao et al. (2020a) showed that those ensemble members that capture the amplitude of the lower stratospheric anomalies during this event (and the 2019 case considered next) were also more successful in forecasting the surface extremes; they also showed
that this was more relevant than whether the model forecasted a split or displacement of the vortex. As discussed above, Kautz et al. (2020) explicitly identified the increased risk of extreme cold over Europe arising from the stratospheric anomalies. This was also the case in the nudging experiments of Knight et al. (2020), who examined the impacts of relaxing the stratospheric flow in seasonal forecasts initialized at the beginning of the winter season. The nudged ensemble reproduced a tropospheric response following the SSW in close agreement with observational composites.

The MJO reached near-record strength in phase 6 and 7 prior to the stratospheric wind reversal in February of 2018, i.e, the MJO phase which has been linked to enhanced SSW frequency and predictability for earlier events (Garfinkel et al., 2012; Garfinkel and Schwartz, 2017). A week after the event the MJO entered phase 8, which is linked to a negative NAO pattern. While Butler et al. (2020) do not find a correlation between forecast errors in the MJO and those in the NAM, Knight et al. (2020) do find that nudging the tropical evolution produces a negative NAO response in late February, suggesting that tropical
circulation anomalies contributed to the anomalous European weather regimes.

The S2S prediction systems forecast the event about 11 days in advance (Karpechko et al., 2018; Rao et al., 2020a), making this event less predictable than some other sudden warmings. Proximately, this is likely due to the nature of the relevant wave

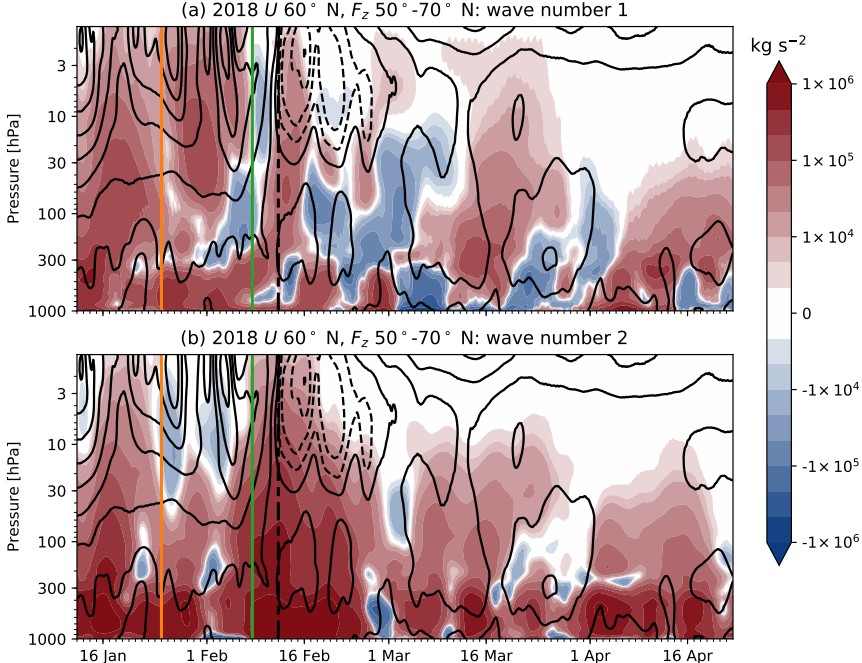

**Figure 6.** Wave forcing during the February 2018 boreal major warming. Shading shows the vertical component of the Eliassen Palm flux averaged from $50°$-$70°$ N from zonal wavenumber (a) one and (b) two. Contour lines show zonal mean zonal wind at $60°$ N; the contour interval is 10 m/s. The vertical black dash-dotted line indicates the date of the wind reversal at 10 hPa, $60°$ N. The green and orange vertical lines indicate the requested initialization dates.

driving which amplified rapidly during the week prior to the stratospheric wind reversal (Fig. 6). Subseasonal forecasts that captured this wave event were more successful in forecasting the vortex breakdown. The difficulty in forecasting the pulse of wave activity has in turn been tied to both anomalous blocking over Siberia (Karpechko et al., 2018) as well as to an episode of anticyclonic Rossby wave breaking in the North Atlantic (Lee et al., 2019).

5    On longer timescales, Knight et al. (2020) further suggest a role for the large-amplitude MJO event that preceded the stratospheric wind reversal, and Lü et al. (2020) suggest that several large snowfall events over Siberia in early and late January contributed to the wave driving responsible for the vortex breakdown. On seasonal timescales, the tropical Pacific was in a moderate La Niña state, and the QBO winds were persistently westerly at 50 hPa and easterly at 30 hPa throughout the winter. Thus the state of both ENSO and the QBO may have also contributed.

10    The first initialization date proposed is 25 Jan 2018, just prior to the first pulse of wave activity leading to the vortex split. The ensembles will thus produce some diversity in the tropospheric precursors outlined above. By considering the ensemble spread, the relative roles of tropospheric precursors and the stratospheric state in the amplification of the planetary waves can thus be isolated. These integrations may also capture some of the development of the European cold air outbreak in late

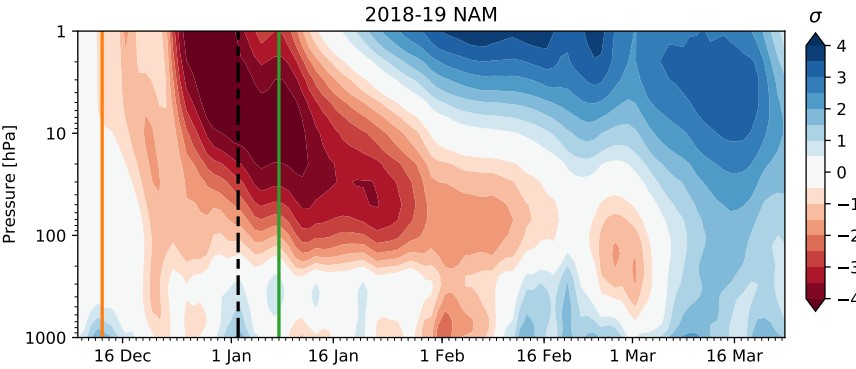

**Figure 7.** NAM indices during the January 2019 boreal major warming. The vertical black dash-dotted line indicates the date of the wind reversal at 10 hPa, 60° N. The green and orange vertical lines indicate the requested initialization dates.

February. The second date, 8 Feb 2018, is chosen to be closer to the development of the tropospheric extreme event, after the full development of the stratospheric anomalies.

### 4.2 Boreal Major Warming of 2 January 2019

In late December 2018, the Arctic vortex was displaced off of the pole, prior to splitting. The 10 hPa winds reversed on 2 Jan
2019. In contrast to the 2018 event, the stratospheric vortex anomalies developed much more gradually through late December and early January of the 2018-19 winter (Lee and Butler, 2020). The vortex remained split for several weeks. Anomalies in the lower stratosphere persisted nearly to March of 2019. The gradual weakening of the vortex was due to persistent wavenumber-one forcing that was well predicted even from mid-December (Rao et al., 2020a).

In strong contrast to the 2018 case, the tropospheric NAM did not respond strongly to the stratospheric anomalies, remaining
near neutral or even slightly positive through much of the troposphere until early February (Fig.7). For instance, the 500 hPa NAM index averaged for one month after the 2018 event was -1.17$\sigma$; following the 2019 case the value was 0.02$\sigma$. However, an extensive cold snap occurred over North America in late January (roughly 23-29 Jan) in a region vertically aligned with one of the daughter vortices generated by the split.

This event was also considered by Rao et al. (2020a), who found that the surface temperatures and precipitation patterns
20 days following the onset date were generally not well forecast by the S2S models. Note, however, that they did not focus specifically on the cold air outbreak over North America. Knight et al. (2020) also performed nudging experiments to explore the impacts of the stratospheric anomalies on the surface. They found that the ensemble mean again reproduced the 'canonical' tropospheric response, with an anomalously persistent negative AO pattern coinciding with NAM anomalies in the lower stratosphere, implying that the lack of tropospheric signal in observations was due to some competing effect(s). One possibility
is that these competing effects arise from the tropics; the tropical nudging experiments of Knight et al. (2020) gave rise to North Atlantic mean sea-level pressure anomalies that more closely resembled observations in January. For instance, the MJO also progressed through phase 6 and 7 in early January 2019, but at amplitudes considerably weaker than in 2018.

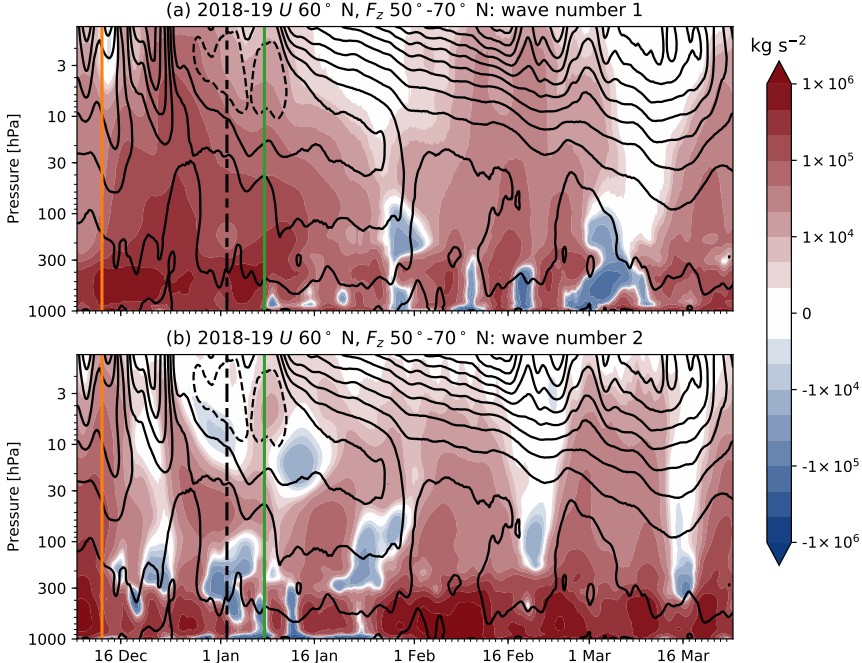

**Figure 8.** Wave forcing during the January 2019 boreal major warming (quantities the same as Fig. 6). The vertical black dash-dotted line indicates the date of the wind reversal at 10 hPa, 60° N. The green and orange vertical lines indicate the requested initialization dates.

The S2S prediction systems forecast the stratospheric wind reversal more than 18 days prior to the central date in some cases (Rao et al., 2020a), but did not predict the vortex would split more than a few days in advance (Butler et al., 2020). The longer forecast horizon in this case seems to be related to the persistent wavenumber-one forcing from mid-December 2018 that displaced the vortex off the pole, prior to its ultimate splitting (see Fig.8).

5    Rao et al. (2020a) propose a range of contributing factors for the wave amplification, including the state of ENSO, the QBO, the solar cycle, and the MJO. In the fall of 2018, the QBO at 50 hPa was strongly easterly, below a westerly shear zone that stretched from 40 hPa to 20 hPa. This shear zone descended through the winter. At the time of the wind reversal, the winds at 50 hPa were easterly and those at 30 hPa were westerly. Since the vertical wind shear in the 30-50 hPa layer was easterly in February 2018 and westerly in January 2019, these two start dates provide contrasting case studies of how model biases in wave forcing and QBO amplitude develop in the lower stratosphere for both easterly and westerly QBO phases.

10    The first suggested initialization date is 13 Dec 2018, just prior to the onset of the wavenumber-one pulse, again motivated by the goal of producing some diversity in the tropospheric wave source in order to distinguish tropospheric and stratospheric contributions to the wave amplification. The second suggested initialization date is 8 Jan 2019, several weeks prior to the North American cold air outbreak.

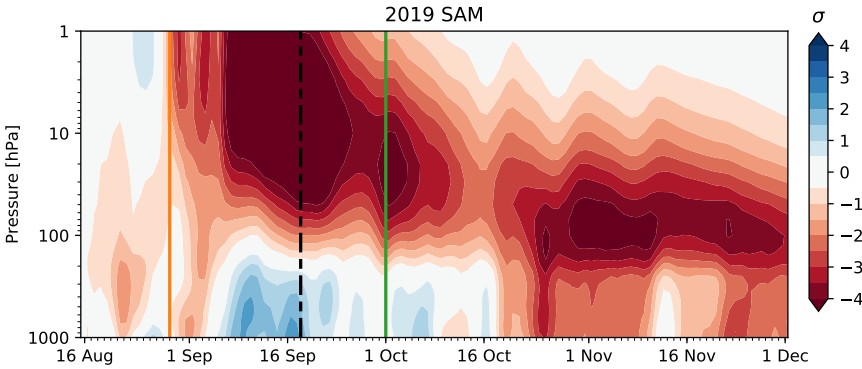

**Figure 9.** SAM indices during the September 2019 austral minor warming. The SAM indices are computed from ERA5 geopotential height anomalies averaged over 65° to 90° S, analogously to the NAM indices shown in Figs. 5 and 7. The vertical black dash-dotted line indicates the date of minimum zonal mean zonal wind at 10 hPa, 60° S. The green and orange vertical lines indicate the requested initialization dates.

## 4.3 Austral Minor Warming of September 2019

The final event of interest is the minor warming that occurred in the Southern Hemisphere in September of 2019. Significant SAM anomalies began to emerge in the upper stratosphere towards the end of August (Fig. 9). However, in contrast to the first two cases, the zonal mean winds at 10 hPa, 60° S did not reverse. However, they did decelerate dramatically, reaching their minimum value on 18 Sep 2019, which can be considered as the 'central' date for the event (Fig. 10). In late August the mid-stratospheric winds were near their climatological values, before a series of wavenumber-one pulses of upward wave activity flux weakened the vortex from the stratopause downwards (Lim et al., 2021, see also Fig. 10).

The tropospheric SAM did not initially shift to negative values following the event. However, negative anomalies were observed in late October and November, during which conditions over Australia were hot and dry; severe wildfires were widespread in November and unprecedented in December, potentially due in part to the stratospheric anomalies (Lim et al., 2019, 2021).

Noguchi et al. (2020) studied the effects of this event on tropical convection though a nudging experiment constraining the full stratospheric flow. They demonstrated that the stratospheric anomalies led to a systematic enhancement of convection in the Northern Hemisphere tropics, centered over Southeast Asia and the Western Pacific.

The event was forecast nearly 18 days prior by models with a reasonably resolved stratosphere (Rao et al., 2020b), including the persistent stratospheric wavenumber-one flux anomalies (Fig. 10). A number of tropospheric precursors have been linked to this wave activity pulse, including a persistent blocking high over the Antarctic Peninsula and a region of anomalously low pressure over the Southern Indian Ocean (Rao et al., 2020b; Lim et al., 2021). The first suggested initialization date is 29 August 2019, early in the development of the wave activity pulse responsible for the stratospheric event. The second suggested initialization date is 1 October 2019, after the stratospheric anomalies are established, two to three weeks prior to the onset of the tropospheric SAM response.

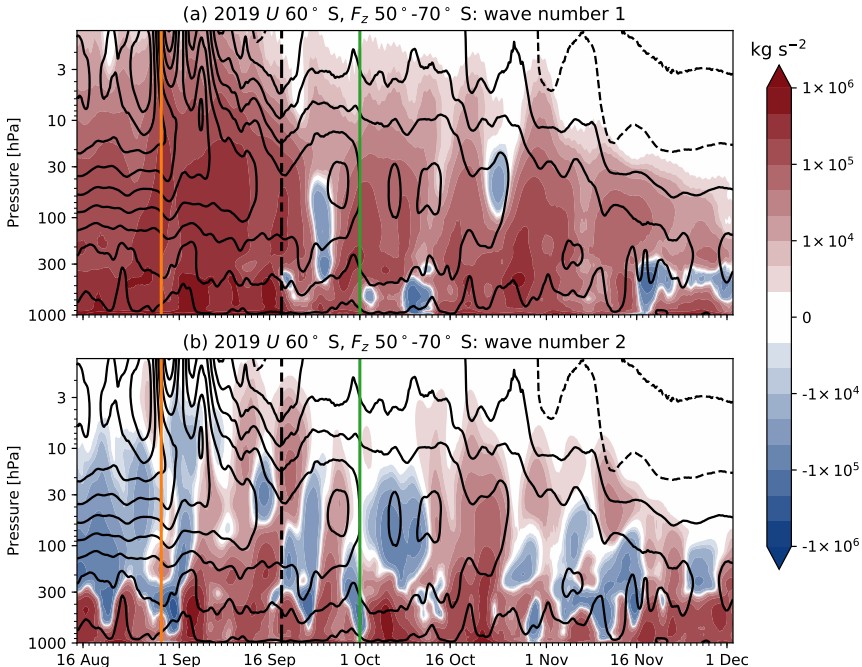

**Figure 10.** Wave forcing during the September 2019 austral minor warming. Shading shows the vertical component of the Eliassen Palm flux averaged from $50°$-$70°$ S from zonal wavenumber (a) one and (b) two. Contour lines show zonal mean zonal wind at $60°$ S. The vertical black dash-dotted line indicates the date of minimum zonal mean zonal wind at 10 hPa, $60°$ S. The green and orange vertical lines indicate the requested initialization dates.

A major SSW occurred in the Southern Hemisphere in September of 2002 during which the polar vortex westerlies did fully reverse. The stratospheric anomalies during the 2019 event were in fact comparable to those observed in 2002, but the latter occurred slightly later in the spring when the climatological westerlies are weaker. We have chosen to focus on the 2019 case instead due to the recent focus on the latter event from an S2S perspective, and because of the connection that has been drawn to the severe fires in Australia.

## 5  Data Request

In order to meet the scientific goals of this project, we request output from the forecast models that includes both surface quantities needed for identifying and quantifying high-impact surface extremes, as well as dynamical quantities needed to diagnose the processes that couple the stratosphere and troposphere. Given the relatively short integration periods and small number of initialization dates, data is requested at relatively high temporal and vertical resolution to enable more detailed comparisons of the relevant processes than has been possible with existing subseasonal forecast databases. The data request is

closely related to the DynVarMIP request (Gerber and Manzini, 2016), including a request for quantities required to close the zonally averaged zonal momentum and thermodynamic budgets.

Quantities are requested on a horizontal grid at no finer resolution than $1° \times 1°$, and at a time resolution of 6 hours. Atmospheric quantities that depend on a vertical coordinate are requested on a slightly non-standard grid "snap34" of pressure levels: 1000, 925, 850, 700, 600, 500, 400, 300, 250, 200, 170, 150, 130, 115, 100, 90, 80, 70, 60, 50, 40, 30, 20, 15, 10, 7, 5, 3, 2, 1.5, 1, 0.7, 0.5, and 0.4 hPa. This includes some additional levels in the lower stratosphere relative to the CMIP6 "plev39" grid, and fewer levels above the stratopause. Variable names follow CMIP6 standard naming conventions where possible. The requested variables are summarized in four tables.

**6hr** Surface quantities and fluxes averaged, maximized, or minimized over the 6 hours preceding the timestep (Table 2).

**6hrZ** Zonally-averaged atmospheric quantities averaged over the 6 hours preceding the timestep (Table 3). Includes quantities needed to close atmospheric momentum and thermodynamic budgets. These follow closely the DynVarMIP request (Gerber and Manzini, 2016), but include the imposed tendencies from the nudging as well.

**6hrPt** Instantaneous basic meteorological and surface quantities output every 6 hours (Table 4).

**6hrPtZ** Instantaneous Transformed Eulerian Mean dynamical quantities output every 6 hours, following the DynVarMIP request (Table 5).

There are two levels of priority for the requested variables. The higher priority variables (1) are considered necessary to meet the primary science goals, and include meteorological quantities (winds, temperatures, specific humidity, and geopotential height) required to compute commonly used dynamical diagnostics, measures of precipitation, and surface quantities including pressure, temperature, and horizontal winds. Variables at the lower priority level (2) include zonally averaged quantities that would permit closing the zonally averaged momentum and thermodynamic budgets, as well as surface quantities that would permit a more detailed analysis of surface processes. The zonal mean stratospheric ozone field is also requested at this lower priority level to allow for some assessment of the importance of ozone anomalies in forecasting the evolution of the polar vortex over subseasonal timescales.

## 6 Summary and Outlook

The SNAPSI project aims to produce a set of controlled ensemble forecasts, initialized around several recent sudden stratospheric warmings. This dataset will allow for an unprecedentedly thorough, multi-model assessment of the contribution of stratospheric extreme events to surface predictability on subseasonal timescales. The proposed forecast ensembles include standard, free-running ensembles, in addition to 'nudged' ensembles in which the evolution of the stratosphere is constrained either to the observed or to climatological conditions. The use of zonally symmetric nudging will enable detailed investigations of the representation of planetary waves that play a central role in the evolution of the events.

**Table 2.** Requested variables in table **6hr**. Surface (XYT) output averaged over 6 hourly intervals. See text for meaning of priority levels.

| Name (Priority) | Long name | Unit |
|---|---|---|
| pr (1) | Precipitation | kg m$^{-2}$ s$^{-1}$ |
| prc (1) | Convective Precipitation | kg m$^{-2}$ s$^{-1}$ |
| clt (2) | Total Cloud Cover Percentage | % |
| hfds (2) | Downward Heat Flux at Sea Water Surface | W m$^{-2}$ |
| tauu (2) | Surface Downward Eastward Wind Stress | Pa |
| tauv (2) | Surface Downward Northward Wind Stress | Pa |
| tasmax$^a$ (2) | 6 hourly Maximum Near-Surface Air Temperature | K |
| tasmin$^a$ (2) | 6 hourly Minimum Near-Surface Air Temperature | K |

$^a$ Maximum/minimum computed over preceding 6 hours.

**Table 3.** Requested variables in table **6hrZ**. Zonal mean output averaged over 6 hourly intervals (YPT). See text for meaning of priority levels.

| Name (Priority) | Long name | Unit |
|---|---|---|
| tntnd$^a$ (1) | Tendency of Air Temperature Due to Imposed Relaxation | K s$^{-1}$ |
| utendnd$^a$ (1) | Tendency of Eastward Wind Due to Imposed Relaxation | m s$^{-2}$ |
| tntmp (2) | Tendency of Air Temperature Due to Model Physics | K s$^{-1}$ |
| tntrl (2) | Tendency of Air Temperature Due to Longwave Radiative Heating | K s$^{-1}$ |
| tntrs (2) | Tendency of Air Temperature Due to Shortwave Radiative Heating | K s$^{-1}$ |
| utendepfd (2) | Tendency of Eastward Wind Due to Eliassen-Palm Flux Divergence | m s$^{-2}$ |
| utendmp$^a$ (2) | Tendency of Eastward Wind Due to Model Physics | m s$^{-2}$ |
| utendnogw (2) | Eastward Acceleration Due to Non-Orographic Gravity Wave Drag | m s$^{-2}$ |
| utendogw (2) | Eastward Acceleration Due to Orographic Gravity Wave Drag | m s$^{-2}$ |
| utendvtem (2) | Tendency of Eastward Wind Due to TEM Northward Advection and Coriolis Term | m s$^{-2}$ |
| utendwtem (2) | Tendency of Eastward Wind Due to TEM Upward Advection | m s$^{-2}$ |
| vtendnogw (2) | Northward Acceleration Due to Non-Orographic Gravity Wave Drag | m s$^{-2}$ |
| vtendogw (2) | Northward Acceleration Due to Orographic Gravity Wave Drag | m s$^{-2}$ |
| xgwdparam (2) | Eastward Gravity Wave Drag | Pa |
| ygwdparam (2) | Northward Gravity Wave Drag | Pa |

$^a$ These variables are not defined by the CMIP 6 standard.

Three case studies have been chosen to apply the experimental methodology: the major boreal sudden stratospheric warmings of February 2018 and January 2019, and the austral minor sudden stratospheric warming of September 2019. The atmosphere

**Table 4.** Requested variables in table '6hrPt'. Instantaneous atmospheric (XYPT) and surface (XYT) quantities output at 6 hourly intervals. See text for meaning of priority levels.

| Name (Priority) | Long name | Unit |
|---|---|---|
| ta (1) | Air Temperature | K |
| ua (1) | Eastward Wind | m s$^{-1}$ |
| va (1) | Northward Wind | m s$^{-1}$ |
| wap (1) | Omega (vertical pressure velocity) | Pa s$^{-1}$ |
| zg (1) | Geopotential Height | m |
| hus (1) | Specific Humidity | kg kg$^{-1}$ |
| ps (1) | Surface Air Pressure | Pa |
| psl (1) | Sea Level Pressure | Pa |
| tas (1) | Near-Surface Air Temperature | K |
| uas (1) | Eastward Near-Surface Wind | m s$^{-1}$ |
| vas (1) | Northward Near-Surface Wind | m s$^{-1}$ |
| rlut (1) | TOA Outgoing Longwave Radiation | W m$^{-2}$ |
| tos (2) | Sea Surface Temperature | degC |
| siconca (2) | Sea-Ice Area Percentage | % |
| sithick (2) | Sea Ice Thickness | m |
| snd (2) | Snow Depth | m |
| snw (2) | Surface Snow Amount | kg m$^{-2}$ |
| mrso (2) | Total Soil Moisture Content | kg m$^{-2}$ |
| mrsos (2) | Moisture in Upper Portion of Soil Column | kg m$^{-2}$ |

**Table 5.** Requested variables in table '6hrPtZ'. Zonal mean, instantaneous atmospheric (YPT) output at 6 hourly intervals. See text for meaning of priority levels.

| Name (Priority) | Long name | Unit |
|---|---|---|
| epfy (2) | Northward Component of the Eliassen-Palm Flux | m$^3$ s$^{-2}$ |
| epfz (2) | Upward Component of the Eliassen-Palm Flux | m$^3$ s$^{-2}$ |
| o3$^a$ (2) | Mole Fraction of $O_3$ | mol mol$^{-1}$ |
| vtem (2) | Transformed Eulerian Mean Northward Wind | m s$^{-1}$ |
| wtem (2) | Transformed Eulerian Mean Upward Wind | m s$^{-1}$ |

$^a$ Climatological if necessary

exhibited a range of tropospheric responses, but in each case, an extreme event with significant societal impacts followed the stratospheric perturbation.

**Table 6.** Participating Centres, models, and reference publications.

| Participating Centre | Model | Reference Publication(s) |
| --- | --- | --- |
| Beijing Climate Center (BCC), China Meteorological Administration | BCC-CSM2-HR | Wu et al. (2019, 2021) |
| Institute of Atmospheric Sciences and Climate of the National Council of Research of Italy (CNR-ISAC) | GLOBO | Malguzzi et al. (2011); Mastrangelo and Malguzzi (2019) |
| Environment and Climate Change Canada (ECCC) | GEM-NEMO | Smith et al. (2018); Lin et al. (2020) |
| Environment and Climate Change Canada (ECCC) | CanESM5 | Swart et al. (2019); Sospedra-Alfonso et al. (2021) |
| European Centre for Mid-range Weather Forecasting (ECMWF) | IFS | ECMWF (2020) |
| Geophysical Fluid Dynamics Laboratory, NOAA (GFDL) | SPEAR | Delworth et al. (2020) |
| Korean Meteorological Administration (KMA) | GloSea5-GC2 | MacLachlan et al. (2014); Williams et al. (2015); Walters et al. (2017) |
| Météo-France | CNRM-CM 6.1 | Voldoire et al. (2019) |
| National Center for Atmospheric Research (NCAR) | CESM2(CAM6) | Danabasoglu et al. (2020); Richter et al. (2021) |
| Naval Research Laboratory (NRL) | NAVGEM | Hogan et al. (2014); McCormack et al. (2017); Eckermann et al. (2018) |
| United Kingdom Met Office (UKMO) | GloSea5 | MacLachlan et al. (2014) |

The experiments have been designed with four primary scientific motivations. First, as outlined above to assess the contribution of the stratosphere to subseasonal forecast skill. Second, to develop methods of formally attributing specific surface extremes to this stratospheric variability. Third, to quantify in detail mechanisms responsible for the surface impacts across the forecast models, controlling for the magnitude and nature of the zonally symmetric stratospheric anomalies that are thought to be most directly responsible for the surface impacts. Fourth, and finally, to improve understanding of the upward coupling from the troposphere to the stratosphere. The experimental design, specific case studies and forecast initialization dates have been chosen to meet these four goals.

Beyond these central goals, the experiments are further expected to shed light on a number of other aspects of dynamical coupling on subseasonal timescales between the stratosphere and troposphere, and between the tropics and extratropics. Notably, both the 2018 and 2019 boreal sudden warming case studies span periods with significant MJO activity and differing

phases of the QBO, and the 2019 austral sudden warming case spans the development phase of a disruption to the QBO that occurred in early 2020.

At the time of submission, eleven modeling groups from ten modeling centers are participating in this project (Table 6). Output from the contributing models will be stored in a central archive hosted by CEDA. Initial analysis of the output will be carried out by community working groups organized through the SNAP project. The output will be made available to the broader community. However, for an initial embargo period, to allow time for this analysis to be carried out, and to ensure modeling centers receive recognition for the time and resources that they have committed, anyone wishing to publish results based on this dataset will be asked to offer co-authorship to the SNAPSI leads and modeling center contacts.

*Data availability.* The reference states for all nudging runs are published and available online through CEDA (Hitchcock, 2022).

*Author contributions.* The protocol was initially designed by Peter Hitchcock, Amy Butler, Andrew Charlton-Perez, Tim Stockdale, and Chaim Garfinkel. James Anstey, Dann Mitchell, and Daniela Domeisen contributed to the text of the paper. The remaining authors contributed to the experimental design from the perspective of contributing operational centers and edited the manuscript.

*Competing interests.* The authors declare no competing interests are present.

*Acknowledgements.* We acknowledge the support of SPARC and the S2S Prediction Project. CIG is supported by the European Research Council starting grant under the European Union's Horizon 2020 research and innovation program (Grant Agreement 677756). DD gratefully acknowledges support from the Swiss National Science Foundation through projects PP00P2_170523 and PP00P2_198896. We thank the Centre for Environmental Data Analysis for archiving the reference data set and the forecasts that will be generated as part of this intercomparison effort.

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
