# Peer review of "Stratospheric Nudging And Predictable Surface Impacts (SNAPSI): A Protocol for Investigating the Role of Stratospheric Polar Vortex Disturbances in Subseasonal to Seasonal Forecasts"

_Geoscientific Model Development, 2021_

## Author Comment (AC1)

*We thank the reviewer for their thoughtful comments. We have replied to each point in italics below. Text in red indicates a change to the manuscript.*

The manuscript outlines a set of protocols for multiple but standardized global climate modeling experiments to study the stratosphere-troposphere coupling under the umbrella called the Stratospheric Nudging And Predictable Surface Impacts (SNAPSI). The authors describe and outline an intercomparison modeling experiment to study the role of the Arctic and Antarctic stratospheric polar vortices in sub-seasonal to seasonal forecast models.

I appreciate that the authors have concerns for a nudging the stratosphere to the full observed state including eddies rather than the proposed zonally symmetric state. Still, I am concerned that only nudging to the zonally symmetric observed state may omit important stratospheric information or forcing on or coupling with the troposphere. I don't have a suggested solution but do want to raise the concern.

*Similar concerns were also raised in RC2, and this was a topic of discussion amongst all of the authors when designing the protocol. There are merits and drawbacks to each approach, and so we discuss here some of the pros and cons to adopting a full-nudging versus a zonal-mean only nudging strategy.*

*There are two primary arguments for full-field nudging: first, that there may be some important role for stratospheric asymmetries in determining the tropospheric response to SSWs, and second, that models operating on a grid that is not aligned with the parallels may be unable to participate in the overall protocol.*

*There are two primary arguments for zonally-symmetric nudging: first, by leaving the wave field to evolve freely, the experiments will allow us to investigate the impact of stratospheric mean-state biases on the forecast of the planetary wave field, and second, that we have a deeper theoretical understanding of the consequences of zonal mean nudging.*

*Past work has demonstrated that much of the surface response is in fact captured by the zonal mean nudging approach alone. Planetary waves in the extratropical stratosphere are suppressed following SSWs, which means zonal asymmetries in the stratosphere are weak. This work has emphasized the time-mean jet shift component of the response rather than the shift in probability of extremes such as cold air outbreaks, so it is possible that the zonal mean nudging will miss some possible impacts arising from the asymmetric component of the stratosphere.*

*It is not clear, however, that full-field nudging will really provide an 'upper bound' on the downward impacts of the stratosphere. Nudging of any kind in the presence of strong balance constraints implies that there will be unintended, remote consequences of including an artificial forcing. Several model studies (Orbe et al. 2017, Chrysanthou et al. 2019) have pointed out poorly-understood dynamical and transport inconsistencies in specified dynamics integrations. Nudging the asymmetric component of the stratosphere will also introduce an effective and highly artificial reflecting layer for any large-scale Rossby waves that are not consistent between*

*the model forecast and the nudged reference state. The induced zonal asymmetries in the stratosphere may also act as an effective stratospheric source of waves. All of which may produce unintended biases in the surface flow; these effects have not been quantified. In contrast, the dynamical artifacts associated with zonally symmetric nudging are better understood (see Hitchcock and Haynes 2014).*

*Arguably, the zonally symmetric evolution of the stratosphere is an easier target for improved forecasts. Particularly following stratospheric sudden warmings, radiative processes dominate the evolution of the polar vortex. The zonally symmetric nudging ensembles might thus be better regarded as a plausible target for forecast models.*

*Finally, it is again arguable that relaxing the stratospheric zonal mean state to climatology provides a more natural control than relaxing the full field in the stratosphere. The zonal mean climatology is likely to be more dynamically consistent with the tropospheric state than the full field climatology, in which the planetary waves (which have long vertical wave lengths) will be constrained to their quasi-stationary climatology.*

*There is indeed one participating model (the GFDL SPEAR model) which cannot easily carry out zonal mean nudging and will only contribute the full nudging runs. However, the remainder of the models can and will carry out the zonally symmetric nudging. Given that several models may turn to more complex grids, it could be difficult to carry out model intercomparisons using zonally symmetric nudging in the future.*

*In summary, there are strong arguments for carrying out both symmetric and full-field nudging forecasts. Both are included in the protocol, and we expect some modeling centers carry out both, enough that we will be able to carry out some detailed comparisons between the two approaches. On balance, the additional science questions that can be addressed with the symmetric nudging approach was felt to be worth prioritizing.*

Some of these arguments have now been added to the text, both in the introduction and in a new subsection at the end of section 3.

In Figure 2, I suggest that the plot of the observations be made consistent with the forecast pots? I found it hard to compare between observations and the forecasts.

We have modified the final panel of this figure so that the observations have the same color scale as the forecasts. We have also added some additional information in the figure caption.

Line 227 Not sure that I agree with the statement: "Comparisons between the nudged and control ensembles will provide a clear means of assessing the stratospheric pathway at play for those teleconnections that are active during the selected case studies." There could be multiple forcings in play and nonlinear interactions that would make attribution complicated.

*Yes, there are likely to be both multiple forcings and nonlinear interactions, and these experiments will not allow us to disentangle every such interaction. However, the experimental protocol provides a simple and causal way to assess the role of the stratospheric mean state in*

*any pathway. For instance, if teleconnections from the tropical Pacific play a role in the extratropical response, and if these depend on the state of the stratosphere (e.g. Domeisen et al. (2015)), they should be active in the nudged ensembles but not in the control ensembles.*

*We are now more explicit about how these comparisons will shed light on this scientific question.*

3.1 Why are only temperature and zonal winds provided from ERA5. I would have thought to include geopotential height and meridional winds as well?

*Only T and U are required to be nudged in the zonal mean; geopotential height is determined by the temperature field and V in the zonal mean is strongly constrained by hydrostatic balance and continuity (see Hitchcock and Haynes 2014).*

Lines 256-258 This is a difficult balance to strike nudging the stratosphere towards observations without throwing the whole model simulation out of whack. I can understand imposing no nudging below 90hPa that accelerates to full nudging at 100hPa but I don't believe that we fully appreciate the importance and role of the lower-stratosphere separate from the mid-stratosphere in stratosphere-troposphere coupling. In fact, I believe that the lower- and mid-stratosphere could influence the troposphere somewhat independently. I am concerned that by imposing now nudging in the lower stratosphere will dampen the full influence of the stratosphere on the troposphere. One idea that I would suggest considering is applying the limit of the nudging to different levels.

*We agree that the question of which levels within the stratosphere are most relevant for stratosphere-troposphere coupling is an interesting and important question, and one that is not fully understood. The lower stratosphere has been shown to be particularly relevant for understanding the impacts of stratospheric sudden warmings (e.g., Karpechko et al. (2018). On the other hand, the mid-stratosphere is thought to be more relevant for planetary wave reflection (e.g., Perlwitz and Harnik 2004). The impacts of imposing nudging at different levels has been considered in detail in a simpler model context by Hitchcock and Haynes (2016), who found that the surface impacts were stronger when the lower stratosphere was better constrained.*

*While this issue certainly warrants further research, our goal here was to constrain as much of the stratosphere as was feasible without directly impacting the troposphere. The choice to ramp up the nudging from 90 hPa to full strength at 50 hPa is similar to the lowest level of nudging considered by Hitchcock and Haynes (2016), remains well above the level of the extratropical tropopause (which is more than a scale height below), and is low enough to constrain the lower stratospheric QBO winds which are thought to be important for their tropical impacts.*

Lines 340-343 – I felt that the discussion about the MJO and its possible influence on the NAM and Northern Hemisphere weather is an unnecessary distraction almost like "having your cake and eating it." The paper is about stratospheric influence and stratospheric nudging so why introduce that the MJO is needed to simulate the correct weather? I think better to leave tropical forcing and guidelines for modeling experiments to study tropical forcing for anther paper.

*There is significant diversity in the surface response to stratospheric sudden warmings. However, a key question in the context of S2S predictability is the origin of this diversity. Does this diversity arise exclusively from synoptic-scale tropospheric processes that may only be predictable for a week or so in advance? In this case this diversity would essentially be `irreducible' on subseasonal timescales for any given event. However, if some of this diversity occurs because of other subseasonal drivers, we may be able to say in advance whether or not a given event will lead to a significant surface response. The results of Knight et al. suggest that the state of the MJO may impact the NAO in late February, which suggests a possible predictable control on the surface impacts.*

*If the diversity is due to unpredictable components, these should differ from ensemble member to ensemble member, and should be independent of nudging imposed in the stratosphere or of initial conditions. If, on the other hand, they are related to other forcings that can be predicted on sub-seasonal timescales, they should be present in ensemble means, but should differ between, e.g. the 2018 and 2019 events, or possibly between different initialization dates for the same events.*

*We do not know a priori which modes are most relevant to this diversity; we quote here the state of various potentially important modes of variability in part for the reference of future studies analyzing output from these runs.*

*We have added text to further clarify why we are quoting the state of these remote climate drivers.*

Lines 361-365 I don't disagree that the tropospheric NAM response in 2019 was quite different than the tropospheric NAM response in 2018 to the stratospheric polar vortex split.  However just by looking at Figures 3 and 5 it is not that obvious to me.  I wonder if a different comparison might better highlight the difference.

*The time-averaged NAM anomaly at 500 hPa for the one-month period following the central date after the 2018 SSW case (12 Feb 2018 through 12 Mar 2018) was -1.17σ. After the 2019 SSW (2 Jan 2019 through 2 Feb 2019) it was 0.02σ. We now quote these values in the text.*

*The composite average response following SSWs is negative, more consistent with the 2018 case. As discussed above, a central question to be investigated is whether this difference is predictable, and whether it can be attributed to any specific remote climate driver.*

Lines 376-379 I agree that the tropospheric response differences to the SSWs in 2018 and 2019 are interesting and is worthy of model experiments.  But again I do question introducing into the discussion the MJO and tropical forcing.  Almost makes the role of the stratosphere seem like noise rather than a signal and therefore could be ignored.  My opinion is to take out this mostly hand wavy discussion of the MJO and tropical forcing, which seems self-defeating in trying to motivate stratosphere-only sensitivity experiments.

*As discussed above, these are potentially relevant climate drivers that may be shaping the details of the response to these specific events. We have left this discussion in, as justified above.*

Line 400 I do winder why the September 2019 Austral minor warming was chosen over the September 2002 major warming? In fact the SAM was much more negative in October 2002 (I believe a record in fact) than 2019. If we assume that a reversal of the winds at 10hPa is necessary for the tropospheric response, how do you justify including a case where the winds never reverse? I do believe that the September 2019 austral polar vortex disruption is interesting but seems like not a good fit for the framework of this study. The first two words in the Abstract are "major disruptions." At a minimum justification of the choice is needed.

*We disagree with the premise that the 10 hPa winds must reverse in order for there to be surface impacts. Minor warming events (in which these winds do not reverse) have been shown to impact the surface (Thompson et al. 2005; Lim et al. 2019, see their Fig. 4). The austral polar vortex was significantly and substantially disturbed throughout the stratosphere in the 2019 case (see Fig. 7) despite the fact that the winds at 10 hPa did not reverse. The anomalies in the 2019 case were in fact of very comparable amplitude to the 2002 case but occurred somewhat earlier in austral spring.*

*The 2002 event would also have been a valuable case study to consider. We chose to focus on the 2019 case for several reasons. Firstly, recent work has focused on this event from an S2S context. The 2002 case has also been highly studied, but not necessarily within the S2S context; moreover there are many open questions about the dynamical mechanisms that triggered the 2019 event and its surface impacts. Secondly, previous work has attributed the hot and dry extremes over Australia to these stratospheric anomalies, making this an interesting case to consider from the point of view of the dynamical attribution of extreme events. This has now been made more explicit in the text.*

*Thompson, D. W. J., Baldwin, M. P. & Solomon, S. Stratosphere–troposphere coupling in the Southern Hemisphere. J. Atmos. Sci. 62, 708–715 (2005).*

Line 471 – I am surprised by the data being embargoed initially. Seems counterproductive to me.

*The purpose of the embargo is simply to ensure that the modeling center participants receive credit and recognition for the resources and efforts that they put into the design, execution and post-processing of the experiments. Anyone interested in analyzing the output will have access to the data from the archive, but will be required to offer co-authorship to the modeling center participants and SNAPSI leads for any paper published within the embargo period. We feel this is a fair request that will not hinder community access to the dataset. We realize this was not made clear in the submitted draft; this has now been clarified.*

---

## Author Comment (AC3)

Review of "Stratospheric Nudging And Predictable Surface Impacts (SNAPSI): A Protocol for Investigating the Role of the Stratospheric Polar Vortex in Subseasonal to Seasonal Forecasts" by Hitchcock et al.

**General comments**

This manuscript describes an experimental protocol for multi-model assessment of the contribution of SSW events to surface predictability on sub-seasonal timescales. By adopting the nudging approach, this experimental protocol aims to reveal the influence from the "perfect" stratosphere explicitly. This experimental plan is coordinated by the SNAP working group of WCRP SPARC, and is a plan that the SNAP should have submitted and undertaken earlier. After the Phase-I multi-model experiment of SNAP (Tripathi et al. 2016), this community spared time for the "coordinated" (or dull self-nominated) analyses of S2S prediction data. However, as explained in sections 1 and 2, these Phase-II data analyses were almost impossible to disentangle the stratospheric influence on the tropospheric forecast skill with confidence in the causal relationship (I knew that before they did). Therefore, this kind of experiment is necessary to advance our understanding of the stratospheric influence on the tropospheric circulation and to build a common view of expectable skill contribution from the stratosphere in current prediction systems. I support the importance of this proposal.

However, as a reviewer, I feel a little concerned about achieving the purpose of multi-model inter-comparisons in the current proposed settings. In particular, the author's preference of the nudging only zonally symmetric component and the inclusion of the fourth purpose (about wave evolving process in the stratosphere) may prevent a sound comparison of tropospheric response to the prescribed stratospheric state among prediction systems. I could not convince the propriety of the settings, at least from this manuscript. Moreover, the treatment of tropical coupling seems to be inappropriate. Therefore, I recommend the authors reorganize the priority of scientific purposes and show the validity of experimental settings.

We thank the reviewer for their insightful comments and suggestions, in particular with regards to the scientific priorities and the design of the nudging. We have responded to these concerns in more detail below. Our comments are in italics; text in red indicates a change to the manuscript. In brief, we have added some preliminary analysis of model output using the given nudging settings to demonstrate the appropriateness of the nudging settings, and have provided further justification for the science priorities we have chosen.

**Major Comments**

(1) Is it possible to present some evidence for the validity of experimental settings?

I believe that the experimental protocol's main purpose is to share the fixed details of the setting after enough validations (which prevents others from laborious processes checking dependency on settings). In addition, the presentation of a typical (prototype) result would facilitate further participation by others (e.g., Held and Suarez (1994) presented results of two dynamical cores, and it helps the readers and following investigators to deduce the robustness of results). Since

this series of experiments depends largely on the nudging parameters, how the authors have fixed the parameters should be explained with enough reasoning. For example,  $p_b$  and  $p_t$  (and function) are different from those of Hitchcock and Simpson (2014). How did you tune these settings? I guess that the authors have conducted test experiments by using some operational system (the IFS?). How significantly affected the choice of a lower limit of the nudging on the tropospheric ensemble spread and mean difference? Proactive presentations of such information would prevent unnecessary future discussions in the step of inter-comparisons.

We have added two new figures showing preliminary output from one participating model (CESM2), along with further discussion of the choice of nudging parameters. These figures show the effects of the zonal symmetric nudging on the ensemble spread of zonal mean zonal winds, meridional heat flux (as a proxy for vertical wave propagation), and tropical temperatures.

(2) Isn't it too greedy to include the fourth purpose?

The zonally-symmetric nudging allows planetary waves to evolve freely (to some extent) even in the stratosphere. This enables the current protocol to address the fourth purpose. However, at the same time, it allows uncertainty of stratospheric state despite that the most important purpose of this experiment is to assess the contributions from the imposed "perfect" stratosphere and compare them among multi-model results. I feel that the well-tailored nudging of full stratospheric state (e.g., middle-to-upper stratospheric full-nudging with a wider buffer zone below) may be more appropriate to pursue the multi-model inter-comparisons. It would be better if the fourth purpose is placed as an additional scientific goal.

A similar concern was raised in RC1, and we refer readers to our response to that comment for a more complete discussion of the relative merits of zonally symmetric versus full-field nudging. In particular, there are good reasons to be concerned about introducing unintended artifacts within the troposphere when nudging to the full field within the stratosphere; these effects have not been quantified by previous work. In contrast the effects of zonally symmetric nudging are better understood, and previous studies have demonstrated that this approach produces much of the expected surface response. Moreover, the fourth science goal will be valuable for understanding limits to the predictability of the SSWs in the first place, which is fundamental for capturing in advance any associated downward impact of the stratosphere. We have added some further discussion of these points in the introduction and in the section on the nudging setup.

Nonetheless, we expect that we will have both zonally symmetric and full-field nudged experiments available to compare the results. The SNAPSI dataset may help to shed light on the tradeoff between the two nudging approaches.

(3) It may be better to change the nudging setting to discuss the tropical coupling.

Although this manuscript roughly touches the coupling between the tropical stratosphere and the troposphere as the secondary science questions (the fifth purpose), this topic has the potential to be a more important target than extratropical coupling. One of the ultimate purposes

of the multi-model inter-comparison is to attribute the model's performance to some particular model settings. As many modelers would agree, one of the most uncertain parts of atmospheric models is the representation of clouds. Therefore, it is natural that stratospheric influence on tropical convections should be placed at the highest priority of multi-model inter-comparisons. In such an investigation, the lower limit of nudging in the tropics should be set higher than that in the extratropics (not to interfere in the high tropical cloud directly). However, I am unsure whether the current setting ( $p_b = 90$  hPa) is high enough to avoid the direct influence. It may be better to introduce latitudinal dependence in the nudging coefficient if the tropical ensemble spread shows an undesirable distribution. Otherwise, it would be better to plan the nudging experiment focusing on the tropical coupling separately. Sloppy spotlighting may ruin chances of further development.

The state of knowledge about extratropical coupling between the stratosphere and troposphere is much more mature than that of stratosphere-troposphere coupling in the tropics. The impacts of stratospheric sudden warmings on the surface are substantial and well-documented by many studies; moreover this methodology is well-established and understood theoretically, as is described in the methodology section. We are in the right position now to carry out this intercomparison exercise for extratropical coupling and our priorities reflect this.

In contrast, efforts to study the impacts of the QBO on MJO with this methodology have been mixed to date. Martin et al. 2021 carried out a similar experiment in a single model study with a range of nudging parameters, including one experiment with the nudging transition set from 100 hPa to 50 hPa; they did not find an MJO connection. In contrast, Noguchi et al. (2020), using full-field nudging but with a considerably higher transition region from 40 hPa to 1 hPa, found substantial impacts from SSWs on tropical convection more broadly. While there is no doubt that understanding tropical coupling between the stratosphere and troposphere is a key research topic, it is not at all clear what the optimal nudging strategy is to capture the details of the somehow affected by the nudging is one aspect to consider. More single model studies are required to understand these kinds of concerns. Moreover, it is likely that this would differ depending on which aspect of coupling in the tropics one is interested in, as is suggested by the contrasting results from the Noguchi et al. (2020) and Martin et al. (2021) studies.

In the absence of clear guidance from previous studies, our approach was to set the lower boundary of the nudging sufficiently low to constrain the state of the QBO in the lower tropical stratosphere without introducing artificial constraints in the extratropical upper troposphere.

Nonetheless, the present experimental design has the potential to reveal aspects of two-way stratosphere-troposphere coupling in the tropics, and we feel it is appropriate to highlight this potential as a set of secondary science questions.

(4) Changing the priorities of the experiment will allow more models to participate.

Among the experiments listed in I.75-89, the "free" and "nudged-full" are free from the artificial relaxation procedure with shocks and relatively easy to conduct even by models with grids that are not necessarily harmonic with the zonally-symmetric nudging. I think it is better to set these

two experiments as the first step request. Then, other experiments ("nudge," "control," and "control-full") should be requested as the second step. Such a division would increase possible participants, at least for the first step. Since the "free" experiments would approach the model's climatological state if the initialization date is set far enough from the SSW onset date (although there are exceptions, of course), the purpose of deducing stratospheric contribution to the troposphere can be roughly achieved by just comparing the "nudged-full" and "free" experiments. I agree that there are large merits of conducting the zonally-asymmetric nudging and comparing it with the "control" experiment. However, I wonder which should we place the priority in the multi-model inter-comparison.

It is true that the nudged-full forecast is easier for models with grids that are not aligned with the parallels. There is one participating model that will carry out the full-field nudging and not the zonally symmetric case. However, it is not at all true that they are free from artifacts associated with the nudging. The boundary between the free troposphere and the nudged stratosphere will act as a strong, unphysical reflecting layer for any large-scale Rossby waves that are inconsistent between the model forecasts and the nudged stratosphere.

**Minor Comments**

Title: the Stratospheric Polar Vortex —> e.g., "Recent Weakening Events" of the Stratospheric Polar Vortex

Since this protocol covers just only 3 SSWs which are mainly touched by recent publications of quick S2S data analyses, it is inappropriate to use the term representative of various behavior of stratospheric polar vortices.

We have changed the title to "Stratospheric Nudging And Predictable Surface Impacts (SNAPSI): A Protocol for Investigating the Role of Stratospheric Polar Vortex Disturbances in Subseasonal to Seasonal Forecasts"

Is the "control-full" setting appropriate?

Unlike the "control" experiment, the "control-full" experiment would strongly damp the stratospheric wave components due to the sample-averaged smooth structure of the climatological state. Is this as you intended? I think the true "control-full" experiment should construct its ensemble by changing  $T_{c}(t)$  to  $T_{year}(t)$  (each year's state of ERA5). In this case, at least a 40-member ensemble can be obtained using the reanalysis data from 1979 to 2018.

Yes, the control-full specification will strongly constrain the stratospheric state. Like the control ensemble, it will provide an assessment of the effects of the zonally asymmetric stratospheric anomalies on the surface relative to a climatological state. It is also true that there will be artifacts associated with constraining the wave field in the stratosphere to something inconsistent with the tropospheric wave field; this is just as true of nudging to the observed stratospheric flow in other years, or of nudging to an observed state that is inconsistent with model dynamics. Using a different year for each ensemble member would also vastly increase

the size of the reference dataset. This is certainly an interesting idea, but it is not clear that it offers clear benefits compared to the technical challenges it introduces.

Figure 1:

Is it possible to arrange this figure as a more straightforward form for this manuscript? I think the histogram of the split SSW is unnecessary (e.g., Figure 11 of Maycock et al. 2020: Removing CTL\_ADJ is more desirable...).

Figure 1 shows the effects of nudging the zonally symmetric component of the stratosphere on the ensemble distribution of NAO. The two nudged ensembles (SSWs and SSWs) are nudged to different events taken from a free running version of the model, much like the present protocol focuses on several specific case studies. Including the results from both nudged ensembles demonstrates that the result is robust across multiple reference cases, which is quite relevant to the present protocol. We added a brief comment to clarify this in the text, though a full discussion seems out of place in the discussion.

Figure 2:

This figure needs to be brushed up. The observation should be changed to the same format as the forecasts. It seems that the temperature anomalies of the forecasts are limited over the land. How many ensemble members are used to plot in each panel?

We have updated the observations panel to have the same color scale as is used in the forecast plots. We have also added text to the caption providing further information about the forecasts; specifically they include 40 ensembles initialized over a span of 10 days (four runs initialized per day). All panels are now masked to emphasize forecast temperatures over the land.

The caption of Figure 3:

In my understanding, Butler et al. (2020) does not describe the calculation method of NAM indices in detail. They have just cited Gerber and Martineau (2018). I do not really like such an inappropriate citation. It is better to write such as "ERA5 version of Figure 5(a) in Butler et al. (2020)."

We have changed the citation to Gerber and Martineau (2018).

Table 6 and Authorship:

I doubt the necessity of Table 6 and authors from operational centers since the numerical integrations are not performed, and any early results are not provided in this manuscript. They have just only expressed the intention to participate. The authorship of these people should generate when the data are submitted and the model settings are described in some data journals (e.g., ESSD?). Therefore, the contribution of these types should be noted in the acknowledgement.

The protocol was developed with significant input and feedback from the modeling center contacts. The choice of events and initial conditions, design of the nudging, data request and scientific priorities were all determined in consultation with these contacts to ensure they were reasonable from a technical point of view and that they were fit for the scientific purpose. Their authorship is well-justified in this protocol description paper.

**Typos, etc.**

I.320: 60 N --> 60° N

**Done.**

Make consistency in the use of abbreviation terms (NAO, SAM, MJO, QBO). For example, I.396 and I. 403 uses "Southern Annular Mode" although the SAM is already defined in I.307. Also, "NAO" is used in I.129- before the "North Atlantic Oscillation" in I.308.

**We have worked to improve our use of acronyms.**

**References**

Gerber, E. P. and Martineau, P. (2018) Quantifying the variability of the annular modes: reanalysis uncertainty vs. sampling uncertainty, *Atmos. Chem. Phys.*, **18**, 17099–17117, https://doi.org/10.5194/acp-18-17099-2018

Held, I. M., and Suarez, M. J. (1994) A proposal for the intercomparison of the dynamical cores of atmospheric general circulation models, *Bull. Amer. Meteorol. Soc.*, **75(10)**, 1825-1830. https://doi.org/10.1175/1520-0477(1994)075<1825:APFTIO>2.0.CO;2

Maycock, A. C., Masukwedza, G. I., Hitchcock, P., and Simpson, I. R. (2020) A regime perspective on the North Atlantic eddy-driven jet response to sudden stratospheric warmings, *J. Climate*, **33(9)**, 3901-3917. https://doi.org/10.1175/JCLI-D-19-0702.1

Tripathi, O. P., Baldwin, M., Charlton-Perez, and co-authors (2016) Examining the predictability of the stratospheric sudden warming of January 2013 using multiple NWP systems, *Mon. Wea. Rev.*, **144(5)**, 1935-1960. https://doi.org/10.1175/MWR-D-15-0010.1

---

## Author Response (AR2)

**Response to Editor's Comments**

Dear authors,

Please, check the comments from the reviewer. A few minor corrections are necessary before your paper is accepted for publication.

*Our response is given below; the requested corrections have been made in the new version of the manuscript. Changes to the manuscript are in red.*

Also, please, pay attention to the data accessibility section:

1. Include the DOI in the section, not simply a citation to Hitchcock (2022).

*We have added a DOI as requested.*

2. Given the amount of data currently declared (in the order of hundreds of GB), please, consider publishing it in Zenodo.

*CEDA offers a level of support more appropriate to the size and use case expected for the data generated by this protocol.*

3. We can not accept restricted access to the data through registration or similar. Therefore, you must release the data under the "Publically available data" mechanism that CEDA provides, not the "Registered user accessible data".

*Thank you for catching this - there was an unintended permissions issue affecting the reference dataset that has now been corrected. It is now accessible as 'publically available data' without the need for registration.*

4. I have tried to download the data using my CEDA account. When I try to download a file, I get an HTTP ERROR 403, access forbidden. Please, as said before, recheck accessibility and make all the data available without any restriction, request for registration, etc.

*We have confirmed that the data is available without restrictions.*

**Response to Anonymous Referee #2's Comments**

General comments

While my recommendation to reorganize the priority of scientific purposes is not reflected, I think the authors have answered and addressed my comments in their revised manuscript as possible. I thank the authors for adding preliminary output from CESM2 (Figures 3 and 4) and discussing the benefits and drawbacks of zonally-symmetric and full-field nudgings (section 3.3). Although I still think the settings of the full-field nudging and its tropical part can be sophisticated if we deprioritize the fourth purpose, I agree that this is the matter that should be

tried anyway rather than disputed at this stage. I wish these aspects are reappraised after the success of inter-model comparisons participated by as many models as possible. Good luck with the project!

*Thanks again to the reviewer for their helpful comments on this paper - we have corrected the technical issues raised below; please see our point by point responses for details. Our responses are in italics, comments corresponding to changes to the text or a figure are in red.*

Minor comments

Figure 3: Please change the labels of vertical axes to the correct ones (T --> U).

*Thank you for catching this - they are now correct.*

Figure 4 caption: It is better to declare the contour intervals of the ensemble means. The same thing can be said of the contour intervals of the reanalysis in Figures 6, 8, and 10.

*We have now specified contour intervals in the captions for Figs. 4 and 6; the intervals are the same in Figs. 8 and 10 as they are in Fig. 6.*

Figure 4: It is better to arrange the units ([m/s], [K m/s]) same as other figures.

*We have made the unit formatting more consistent.*

l.295-: How many ensemble members are used to make Figures 3 and 4? Is it a 21-member ensemble (like a real-time forecast described in Richter et al.)? or over 50 members as requested in this protocol?

*50 ensemble members are used; this is now stated in the text.*

l.306-310: I think what should be argued in Figure 4 is how the ensemble spread in the tropospheric zonal-mean zonal wind varies among the experiments. The spread below the middle troposphere is not so different between the free and nudged (& control) runs, although there are some shrinks in the nudged (& control) run above 300 hPa.

*The intent of this figure is to show that the nudging is working as intended: that is, that the zonal mean state of the stratosphere has been strongly constrained while the wave field is allowed to evolve freely. How this affects the tropospheric forecast is a central question to be addressed in the analysis of the output.*